# The role of turbulence and internal waves in the structure and evolution of a near-field river plume

Rebecca A. McPherson[1], Craig L. Stevens[1,2], Joanne M. O'Callaghan[2], Andrew J. Lucas[3], and Jonathan D. Nash[4]

[1]Department of Physics, University of Auckland, Auckland, New Zealand
[2]National Institute of Water and Atmospheric Research, Wellington, New Zealand
[3]Scripps Institution of Oceanography, University of California San Diego, USA
[4]College of Earth, Oceans and Atmospheric Sciences, Oregon State University, USA

**Correspondence:** Rebecca A. McPherson (rmcp393@aucklanduni.ac.nz)

**Abstract.** An along-channel momentum budget is quantified in the near-field plume region of a controlled river flow entering Doubtful Sound, New Zealand. Observations include highly resolved density, velocity and turbulence, enabling a momentum budget to be constructed over a control volume. Estimates of internal stress ($\tau$) were made from direct measurements of turbulence dissipation rates ($\epsilon$) using vertical microstructure profiles. High flow speeds of the surface plume over $2 \mathrm{\ ms^{-1}}$ and strong stratification ($N^2 \sim 10^{-1} \mathrm{\ s^{-2}}$) resulted in enhanced turbulence dissipation rates ($\epsilon > 10^{-3} \mathrm{\ Wkg^{-1}}$) and internal stress ($\tau > 10^{-2} \mathrm{\ m^2s^{-2}}$) at the base of surface layer. Internal waves were observed propagating along the base of the plume, potentially released subsequent to a hydraulic jump in the intial $1 \mathrm{\ km}$ downstream of the plume discharge point. The momentum flux divergence of these internal waves suggests that almost $15\%$ of the total plume momentum can be transported out of the system by wave radiation, therefore playing a crucial role in the redistribution of momentum within the near-field plume. Observations illustrate that the evolution of the momentum budget components vary between the distinct surface plume layer and the turbulent, shear-stratified interfacial layer. Within the surface plume, a momentum balance was achieved. The dynamical balance demonstrates that the deceleration of the plume, driven by along-channel advection, is controlled by turbulence stress from the plume discharge point to as far as $3 \mathrm{\ km}$ downstream. In the interfacial layer however, the momentum equation was dominated by the turbulence stress term and the balance was not closed. The redistribution of momentum within the shear-stratified layer by internal wave radiation and other hydraulic features could account for the discrepancy in the budget.

*Copyright statement.* TEXT

## 1 Introduction

The fate of the freshwater, terrigenous material and energy injected into the coastal ocean by river plumes is determined by physical processes close to the river mouth (McCabe et al., 2009). After the initial discharge at the source, the momentum-dominated jet-like inflow evolves into a buoyancy-forced plume in the near-field region (Hetland, 2005). In this near-field

region, the structure and behaviour of the plume are determined by a balance of governing plume dynamics, dominated by lateral spreading and vertical mixing (Hetland, 2012; MacDonald and Chen, 2004). The strong density gradient between the freshwater plume and ambient coastal water results in a buoyancy-driven lateral spreading of the plume (Hetland, 2012). The surface plume layer thins and accelerates which enhances shear at the plume base, leading to shear-instabilities and turbulence.

However, the mixing of low-momentum, high density ambient water into the plume decelerates the flow and reduces shear which leads to a decrease in the density anomaly, thus slowing lateral spreading (Kilcher et al., 2012; MacDonald and Chen, 2004). The plume then propagates through the mid-field region where inflow momentum is lost before transitioning into the far-field. An understanding of the interplay between these near-field dynamics is therefore necessary to characterise local plume behaviour, determine plume evolution and understand the implications for the larger coastal ocean.

In order to evaluate the spatial evolution of the plume, measured quantities are connected to plume dynamics by constructing a momentum budget over a defined finite region of the flow field, termed a control volume. The control volume method has been applied to riverine systems to estimate turbulent transport quantities (MacDonald and Geyer, 2004; Chen and MacDonald, 2006), examine plume spreading (MacDonald et al., 2007), and determine the role of mixing in the near-field plume region (McCabe et al., 2008; MacDonald et al., 2013). The role of turbulence in influencing plume structure is generally examined by

using control volume methods to estimate internal turbulence stress ($\tau$) either indirectly as a residual of the budget components (MacDonald and Geyer, 2004) or directly using microstructure profiles (Kilcher et al., 2012). While reasonable agreement between estimates of turbulence dissipation rates derived from control volume residuals and direct measurements from shear probes has been observed (MacDonald et al., 2007), Kilcher et al. (2012) found discrepancies between indirectly inferred and directly measured $\tau$ using an extensive data set, which indicates either under-sampling of high-stress regions or errors in the

control volume method assumptions. In the present study, $\tau$ is derived from vertical microstructure profiles.

The turbulence in the near-field region of the river plume system studied in this paper was investigated by McPherson et al. (2019) using direct turbulence measurements from microstructure profilers to examine the drivers of turbulent kinetic energy (TKE) dissipation rate ($\epsilon$) variability within the plume. In the initial 0.5 km downstream of the plume discharge point, measurements of $\epsilon$ in the surface plume layer were amongst the highest recorded in oceanic shear flows (maximum $\epsilon > 10^{-2}$

$\mathrm{Wkg}^{-1}$), with $\epsilon$ decreasing with distance from the source. To illustrate the impact of these enhanced rates of turbulent mixing on the overall structure and behaviour of the plume, and on the balance of other governing near-field plume dynamics, a momentum budget over the region can be constructed.

Internal waves can impact the balance of plume dynamics as they carry mass and energy away from the plume front, and facilitate vertical mixing offshore (Pan and Jay, 2009; Kilcher et al., 2010). Internal hydraulics are classified by the internal

Froude number ($Fr_i = u_s/c$), where $u_s$ is the surface speed and $c = \sqrt{g'h}$ is the internal wave speed, $g'$ is the reduced gravity and $h$ is the depth of the surface layer, which sets the criterion for free wave propagation at a plume front. Internal waves are released when the flow transitions from a supercritical ($Fr_i > 1$) to a sub-critical ($Fr_i < 1$) flow regime (Jay et al., 2009). For example, in the Columbia River, packets of internal waves were released from the plume front when $Fr_i$ decreased below unity (Nash and Moum, 2005), and have been observed to carry up to 70 % of the total energy out beyond the Columbia River's

boundaries (Pan and Jay, 2009), extending the influence of the plume far beyond its bounding front. The transition from

supercritical to sub-critical flow can also induce a hydraulic jump which, due to the need to conserve momentum, is highly dissipative (Weber, 2001) and can subsequently release internal waves (Osadchiev, 2018). This conservation of momentum necessarily creates a system that has dissipative losses, thus constructing a momentum balance is the key to understanding the mixing and energy dissipation.

A useful system in which to examine these mechanics are fjords because the deep, narrow basin acts as a large-scale natural laboratory. A riverine inflow discharged into the head of the fjord (Pickard and Stanton, 1980) provides an idealised domain in which to isolate specific aspects of near-field plume dynamics. While the deep bathymetry minimises tidal exchange and the inner basin is sheltered from large ocean swell, fjord-river interactions can been directly applied to coastal plumes as the two systems share many common features: freshwater inflow to the fjord produces density gradients similar to those observed

in major rivers such as the Hudson and Columbia Rivers (MacDonald and Geyer, 2004; Hunter et al., 2010); comparable estimates of $\epsilon$ occur in each of the near-field plume regions (MacDonald et al., 2007; McPherson et al., 2019); and time-variable discharge rates, common to both systems, similarly impact the plume structure and energetics (Yankovsky, 2001; O'Callaghan and Stevens, 2015).

The objective of this paper is to quantify the along-channel momentum budget in the near-field region of a river plume using

high-resolution observations of velocity, density and turbulence. A control volume is used to relate the measured quantities to the budget components, and the balance of the budget and related plume dynamics within the surface plume, shear-stratified interfacial layer and ambient below are examined. The effect of the dynamical balance on plume structure and behaviour is also described. This includes quantifying the role that internal waves and internal hydraulic jumps play in the distribution of energy within the near-field region.

## 75   2   Field setting and Data

A controlled freshwater discharge is carried from alpine Lakes Manapouri and Te Anau through the Manapouri hydroelectric power station and, via a constructed channel, into the head of Doubtful Sound, located on the south-west coast of New Zealand ($45.3\,°$ S, $167\,°$ E) (Fig. 1a,b). Freshwater is discharged at an average flow rate of $Q = 420\ \mathrm{m^3 s^{-1}}$ with maximum $Q = 550$ $\mathrm{m^3 s^{-1}}$, making it the third largest river flow in New Zealand (Bowman et al., 1999). Maximum plume speeds, which exceed 2

$\mathrm{ms^{-1}}$, are comparable to the peak ebb outflow velocity of larger river systems such as the Columbia River ($Q > 7,000\,\mathrm{m^3 s^{-1}}$) (McCabe et al., 2008).

The main fjord is approximately 35 km long, typically less than 1 km wide and has a maximum depth of 450 m south of Secretary Island (Fig. 1b). The freshwater tailrace is discharged into the head of the inner fjord, Deep Cove (Fig. 1c). Deep Cove is 3.6 km long and, flanked by steep topography, has a maximum depth of 126 m that occurs within 50 m of the shoreline. The

exposure of the Fiordland region to prevailing Westerly weather systems and orographic precipitation compounds in annual rainfall in excess of 7 m (Bowman et al., 1999). The tides are predominantly semidiurnal with ranges of 1.5 m and 2.5 m for neap and spring tides respectively (Walters et al., 2001). However, the headwaters of the fjord absorb the momentum of tidal oscillations (O'Callaghan and Stevens, 2015) thus tides do not influence near-field mixing (McPherson et al., 2019). Observa-

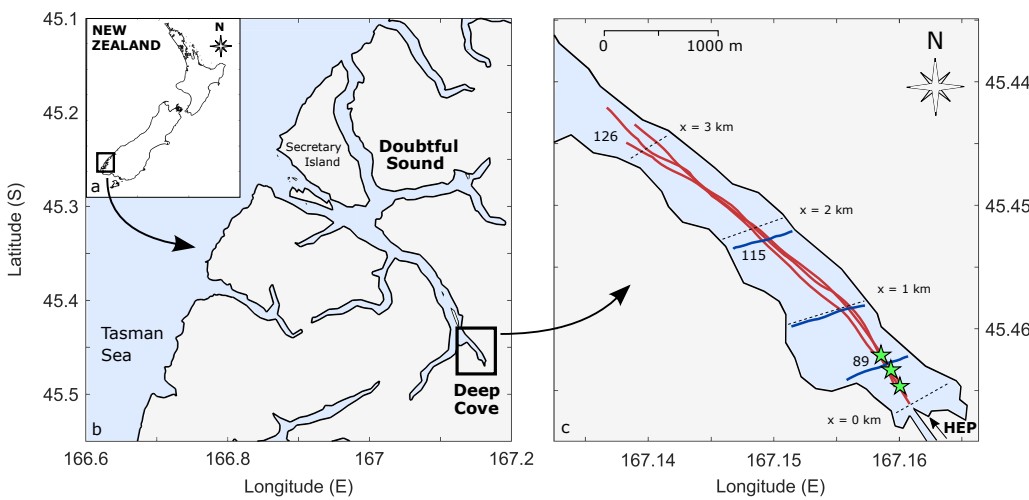

**Figure 1.** Location map of (a) New Zealand with the Fiordland region highlighted, (b) Doubtful Sound identifying the location of (c) Deep Cove showing the repeated across-channel (blue) and along-channel (red) vessel transects taken on three separate days in March 2016. The sampling started at the tailrace exit point and continued downstream towards the seaward end of Deep Cove. The mooring locations (stars) are shown. The arrow at the bottom right of (c) indicates the tailrace inflow from the Manapouri hydroelectric power station (HEP) discharged into the head of Deep Cove. The dashed lines across the fjord are reference distances from the tailrace discharge point ($x = 0, 1, 2, 3$ km) and are referenced in the text and in other figures.

tional experiments were conducted between the tailrace discharge point and the seaward end of Deep Cove, approximately 3 km downstream (Fig. 1c).

A two-week field campaign was conducted in March 2016 when tailrace inflow rates were high and relatively steady ($Q \approx$ 530 $\mathrm{m}^3\mathrm{s}^{-1}$) and wind speeds were typically less than 10 $\mathrm{ms}^{-1}$ (Fig. 2). Though wind mixing generates variability in river plumes (Kakoulaki et al., 2014), it has the greatest effect on plume structure in the far-field (Hetland, 2005) thus wind effects shall not be examined here.

## 2.1 Moored timeseries data

A near-surface instrumented mooring array was deployed in the near-field region of Deep Cove (Fig. 1c). The location and configuration of the moorings were determined by results from previous field campaigns in Deep Cove (O'Callaghan and Stevens, 2015; McPherson et al., 2019). Near-surface velocity was measured by Acoustic Doppler Velocimeters (Nortek Vector) mounted on surface buoys orientated into the seawards-flowing plume and sampled continuously at 1 Hz. The moorings also included temperature loggers (SBE56) sampling at 1 Hz, conductivity-temperature-depth loggers (CTD, SBE37) which sampled every 10 seconds, and an upwards-facing Acoustic Doppler Current Profiler (ADCP; RDI Workhorse 600 kHz) set to record an ensemble every 3 minutes in 2 m vertical bins.

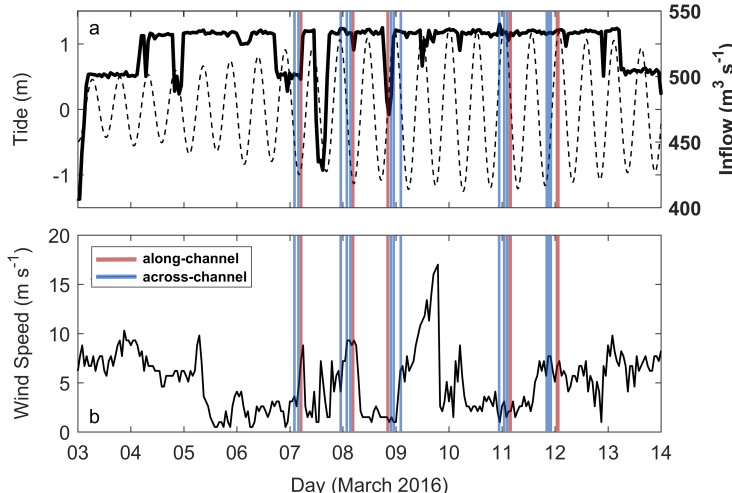

**Figure 2.** Boundary conditions for the duration of the March 2016 experiment. (a) Discharge from the Manapouri hydroelectric power station (solid) and tides (dashed), and (b) wind speed recorded at the head of Deep Cove. Periods when along-channel (red) and across-channel (blue) transects were sampled are indicated.

## 2.2  Vessel-based survey of currents and density field

Along-channel and across-channel vessel transects, aligned with, and perpendicular to, the main river discharge respectively,
were repeated over the course of the field campaign (Fig. 1c, 2). The along-channel transects represented the path of the mean flow as the vessel drifted with the seawards-propagating plume. Data were collected to obtain a spatial distribution of density, velocity and turbulence fields within Deep Cove (Table 1). A weighted bowchain attached to the vessel was comprised of continuously sampling temperature (RBRsolo) and CTD loggers (RBRconcerto), and high-resolution profiles of practical salinity and temperature were obtained from 'tow-yoed' CTD loggers (RBRconcerto) (Fig. 3). These data enabled estimation
of the buoyancy frequency from the gradient of the measured density profiles,

$$N = \sqrt{-\frac{g}{\rho}\frac{\partial \rho}{\partial z}} \tag{1}$$

where $\rho$ is the potential density.

A microstructure profiler (VMP 250, Rockland Scientific) was deployed from the side of the vessel, measuring small-scale velocity shear from which estimates of TKE dissipation rates ($\epsilon$) were directly obtained. The VMP was deployed in an upwards-profiling mode which enabled measurements right to the water surface. Due to contamination of data by the wake of
the instrument when it was released at $\sim 10$ m, measurements of $\epsilon$ towards the bottom of each profile were not always obtained. The sharp velocity gradient between the fast-flowing surface plume and the quiescent ambient below caused the instrument

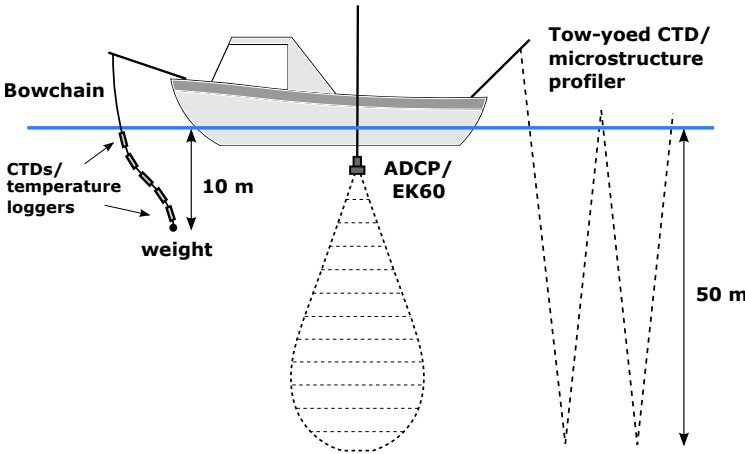

**Figure 3.** Schematic of vessel-mounted instrumentation set-up including bowchain, pole-mounted ADCP and EK60, tow-yoed CTD and microstructure profiler.

to tilt relative to the vertical ($\theta_x$) as it rose towards the surface, and data has been filtered to remove all measurements when $\theta_x > 20°$ (Lueck et al., 2013). Further details about the calculation of $\epsilon$ from velocity shear and other details pertaining to the microstructure data set, including the influence of profiler tilt on the calculation of dissipation rates, are thoroughly discussed
in Appendix A of McPherson et al. (2019).

Horizontal velocity estimates were obtained from a 600 kHz ADCP (RDI Workhorse) mounted on a pole alongside the vessel 1 m below the surface (Fig. 3). Currents were rotated according to the local bathymetry to determine along-channel ($u$) and across-channel ($v$) velocities. Velocity profiles were generally straight from the base of the plume to 2.5 m (Fig. 5b), thus near-surface velocities were obtained by applying a linear fit to the velocity data to extrapolate from 2.5 m to the surface.
The extrapolated velocity profiles were in good agreement with velocity measurements from the velocimeters moored at 0.2 m. Both measured and extrapolated near-surface velocity also compared well to surface currents derived from a series of Lagrangian GPS drifter experiments, in which a pack of surface drifters, released at the tailrace discharge point, were advected with the mean plume flow for approximately one hour ($\sim 3$ km downstream). Furthermore, in-situ velocity measurements up to 1.25 m were obtained from previous field campaigns (McPherson et al., 2019) and good agreement was found between the
linear fit of the extrapolated data from 1.25 m to the surface and the measured velocity in the surface layer.

The internal Froude number, $Fr_i = u_f/c$ is defined using the vessel-based instrumentation, where $u_f$ is the near-surface flow velocity estimated from the ADCP, and $c = \sqrt{g'H}$, where $H$ is the thickness of the surface layer defined by the depth of maximum stratification. This definition of $Fr_i$ has been used in previous river plume studies to determine flow regimes (Hetland, 2012; O'Callaghan and Stevens, 2015; Osadchiev, 2018).

**Table 1.** Summary of Vessel-Mounted Instrumentation[a]

| Instrument | Sampling Rate | Depth Range (m) |
|---|---|---|
| Tow-Yo CTD | 5 Hz | 0 - 50 |
| Bowchain CTD | 5 Hz | 0.5 - 10 |
| Bowchain temperature logger | 2 Hz | 0.5 - 10 |
| ADCP | 600 kHz | 2.5 - 41.4 |
| EK60 | 600 kHz | 0.5 - 38.5 |
| VMP-250 | 512 Hz | 0 - 10 |

**Table 2.** [a]The tow-yoed CTD was continuously profiled with a fall rate of $\sim 1$ ms$^{-1}$. The bowchain was comprised of temperature loggers and CTDs spaced 0.5 m apart. The 600 kHz ADCP was set to sample water velocity continuously in 1 m vertical bins and the 600 kHz echosounder (EK60) was mounted to measure below the surface 0.5 m which was contaminated by vessel motion.

The vertical momentum flux for internal gravity waves can be calculated,

$$F_z = < u'w' > \tag{2}$$

where $u'$ and $w'$ are the horizontal and vertical wave velocities respectively, i.e., the perturbation components after removing the mean flow. The horizontal velocity component ($u$) is from the vessel-mounted ADCP and the vertical velocity component ($w$) is calculated using a control volume method (Eqn. 8) that is described in section 2.3. The mean velocity profiles are calculated from all profiles over the total 3 km length of Deep Cove.

A 600 kHz narrow-beam echosounder (EK60) was also mounted on the other side of the vessel to provide a means of imaging the flow on fine horizontal and vertical scales. The EK60 was positioned 0.5 m below the water surface and measured backscatter in 4.5 cm bins down to 38.5 m (Table 1). Precision position data was obtained from an onboard GPS unit.

## 2.3 Plume momentum

By applying the Boussinesq approximation, the along-channel momentum budget is,

$$\frac{\partial u}{\partial t} + \boldsymbol{u} \cdot \nabla u - fv + \frac{\partial P}{\partial x} + \frac{\partial \tau}{\partial z} = 0 \tag{3}$$

where $f$ is the Coriolis parameter, $\tau$ is the horizontal shear stress and $P$ is the reduced pressure which is written as a sum of its baroclinic ($P_{bc}$) and barotropic ($P_\eta$) components,

$$P = P_{bc} + P_\eta = \frac{g}{\rho_0} \int\limits_z^0 \rho dz + g\eta \tag{4}$$

and $\eta$ is the surface displacement. The observations in Table 1 directly resolved most of the terms in Eq. (3). The local acceleration was calculated as the observed rate of change of velocity between consecutive tow-yoed profiles. Within the advection term, the $u$-component was measured directly using the along-channel velocity, the $v$-component was assumed to be equal to zero along the plume axis, and the $w$-component was obtained using a control volume method. The total plume deceleration is then defined as $Du/Dt = \partial u/\partial t + u\frac{\partial u}{\partial x} + w\frac{\partial u}{\partial z}$. The contribution from the Coriolis force was estimated using the northward velocity component and the baroclinic pressure gradient ($\partial P_{bc}/\partial x$) was calculated from the observed density field. The horizontal shear stress ($\tau$) is,

$$\tau = -K_\nu \frac{\partial u}{\partial z}. \tag{5}$$

and was derived from the vertical eddy viscosity,

$$K_\nu = \frac{1}{1-R_f}\frac{\epsilon}{S^2} \tag{6}$$

where $S$ is the velocity shear and a constant flux Richardson number $R_f = 0.17$ was assumed (MacDonald and Geyer, 2004). In this study, turbulence stress was estimated directly using measurements of $\epsilon$ from the vertical microstructure profiles. This is a development of the control volume technique established by MacDonald and Geyer (2004) which indirectly calculated turbulence as a residual of the momentum budget. Using a large sample size of direct measurements of $\epsilon$ more accurately represents the intermittent and heterogeneous turbulent field, compared to the control volume which provides an integrated effect of turbulent mixing.

The remaining components of Eq. (3) were estimated using a control volume method. As the majority of energy was found in the surface layer (O'Callaghan and Stevens, 2015; McPherson et al., 2019), the upper and lower integration limits were defined by the depth $0 \leq z \leq 10$ m and along-axis distance $0 < x \leq 3$ km. Across-fjord limits (in the $y$-direction) were defined by the relative plume width ($b(x)$) determined by the freshwater conservation equation,

$$Q_0 = \int\limits_z^0 u \frac{s_0 - s}{s_0} b\, dz \tag{7}$$

where the total freshwater flux ($Q_0$) is constant, $s_0$ is the salinity of the ambient and $s$ is the local salinity. The physical characteristics of the plume itself (e.g., $u$, $N^2$, $\epsilon$, etc.) were determined by averaging over the depth of the surface layer ($h$), defined as the distance from $z = 0$ to the depth of the maximum $N^2$ value for each profile. To fully quantify the advection term in Eq. (3), the vertical entrainment velocity ($w$) was calculated using the volume conservation equation,

$$w(z) = \frac{1}{b}\left\langle \frac{\partial}{\partial x} \int\limits_z^0 bu\, dz \right\rangle \tag{8}$$

assuming that $b$ does not vary in time. The remaining barotropic component of the pressure gradient was estimated by calculating $\eta$ using a Bernoulli equation to incorporate the effects of mixing (MacDonald and Geyer, 2004),

$$\eta = \frac{1}{2g}u^2 + \frac{1}{g}\int_0^x \frac{\partial u}{\partial t}dx. \tag{9}$$

using the interpolated surface velocities from the Velocimeter at $z = 0.1$ m.

## 3 Results

### 3.1 Near-field water column structure

The complex vertical structure of the upper water column in the near-field plume region of Deep Cove is illustrated by an along-channel echosounder transect. The bright acoustic scattering layers result from variations in density stratification and microstructure due to turbulence. The transect shows a distinct 5 m deep surface layer in which the scattering intensity is high above a relatively quiescent ambient (Fig. 4). Internal waves can also be seen propagating seawards along the base of the surface layer at 5 m and were not observed to break over the length of Deep Cove.

Vertical profiles relate the sounder observations to the physical characteristics of the upper water column. The high-intensity surface layer observed in Fig. 4 consisted of a $2-3$ m thick freshwater layer ($\sigma_t \approx 3$ kgm$^{-3}$) flowing at speeds of over 1 ms$^{-1}$ above a sharp density interface (Fig. 5a,b). Below the pycnocline, in the low intensity backscatter region, the stationary ambient was of an oceanic density ($\sigma_t = 25$ kgm$^{-3}$) and well-mixed. The strong backscatter at the base of the surface layer was the salinity-induced stratification in the pycnocline ($N^2 = 10^{-1}$ s$^{-2}$), and generally weaker $N^2$ values were observed within the plume layer ($N^2 = 10^{-2}$ s$^{-2}$), reducing towards $10^{-4}$ s$^{-2}$ below the interface (Fig. 5c). In the sounder transect, the braided structures approximately 3 m below the water surface with amplitudes between $0.5-1$ m (Fig. 4, highlighted box) are indicative of shear instabilities, which are a familiar feature in highly stratified shear flows (Tedford et al., 2009; Geyer et al., 2017). These instabilities generated enhanced turbulence at the base of the plume where the billows sustained TKE dissipation rates greater than $\epsilon = 10^{-3}$ Wkg$^{-1}$ close to the discharge point (Fig. 5d).

The plume spread laterally as it propagated downstream, increasing in width from 105 m at the tailrace discharge point to 240 m towards the seaward end of Deep Cove ($db/dx = 0.045$ m$^{-1}$) (Fig. 6a). The thickness of the surface layer varied spatially, fluctuating regularly by 0.5 m vertically over 100 m longitudinally, though $h$ generally decreased in thickness from 5 m at 1 km to 3.6 m at 3 km downstream (Fig. 6b). The velocity of the plume decreased from $1.1-0.75$ ms$^{-1}$ over the 3 km (Fig. 6c) and surface layer $\epsilon$ and $\tau$ also decreased over the length of Deep Cove (Fig. 6e,f). The internal Froude number shows a sharp decrease in $Fr_i$ between approximately 0.5 - 1 km downstream from the tailrace discharge point, where the initially supercritical flow ($Fr_i > 1$) transitioned to a sub-critical flow regime ($Fr_i < 1$) (Fig. 6g).

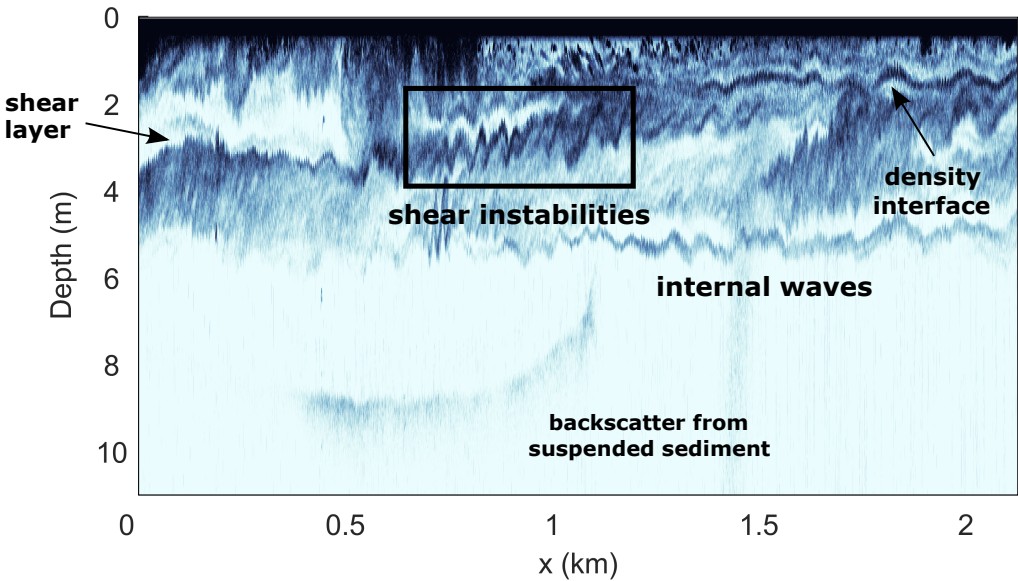

**Figure 4.** Acoustic backscatter intensity from the vessel-mounted echosounder as a function of distance from the tailrace discharge point. The image is coloured so that dark indicates high intensity and white indicates low intensity. The depth of the water column increases from 5 m at the end of the tailrace channel ($x = 0$) to over 90 m in depth over the first 100 m from the tailrace discharge point. The location within the fjord corresponding to the distance axis is shown in Fig. 1c.

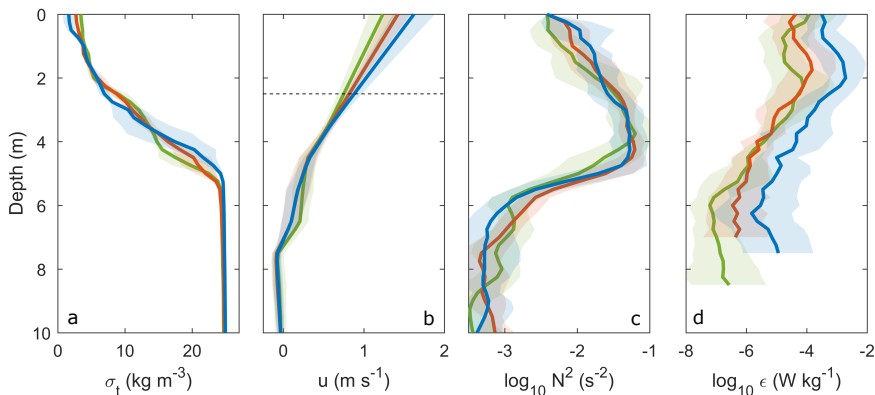

**Figure 5.** Vertical profiles of average (a) sigma-t ($\sigma_t$), (b) along-channel velocity ($u$), (c) stratification ($N^2$), and (d) turbulence dissipation rate ($\epsilon$), averaged over $0 < x \leq 1$ km (blue), $1 \leq x < 2$ km (orange) and $2 \leq x < 3$ km (green). The horizontal dashed line in (b) shows the depth above which the velocity was interpolated. The shading around each profile represents one standard deviation.

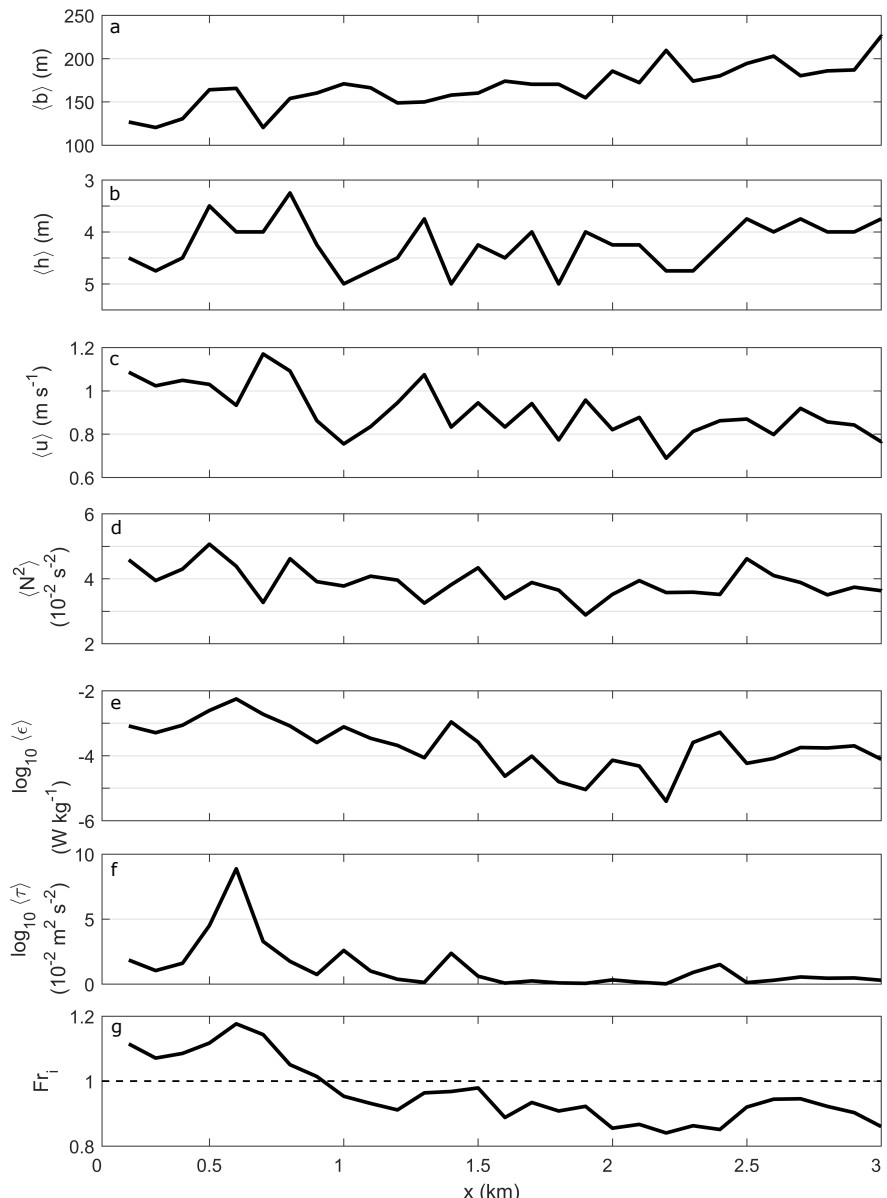

**Figure 6.** The evolution of plume-averaged quantities with distance from tailrace discharge point. (a) Plume width $\langle b \rangle$, (b) surface layer thickness $\langle h \rangle$, (c) along-channel velocity $\langle u \rangle$, (d) stratification $\langle N^2 \rangle$, (e) dissipation rate $\langle \epsilon \rangle$, (f) internal turbulence stress $\langle \tau \rangle$ and (g) internal Froude number ($Fr_i$). The dashed line in (g) is the critical value $Fr_i = 1$. Each term was averaged over the depth of the surface layer ($h$), defined as the distance between the surface and depth of the maximum $N^2$ value. These values are from one along-channel transect on 07 March 2016 and representative of the plume behaviour observed during all transects.

## 3.2 Momentum balance evolution

The individual components of the momentum budget (Eq. (3)) were averaged over three $1$ km long sections from the tailrace discharge point to the seaward end of Deep Cove (see Fig. 1c for reference locations). Nearest the tailrace discharge point ($0 \leq x < 1$ km), the along-channel component of advection in the surface layer and a weaker local acceleration term (Fig. 7b) drove strong plume deceleration in the surface layer ($Du/Dt < 0$, Fig. 7c). The error bounds on the advection components relate to the variability in near-surface flow speeds (Fig. 6c). Below the plume at $4$ m, $Du/Dt$ tended to zero and the ambient was relatively steady. The spreading component was weak within the plume as $w$ was defined to be zero at the surface (Fig. 7b).

The mean pressure gradient ($\partial P / \partial x$) was the same sign and approximately half the magnitude of $Du/Dt$ within the surface layer (Fig. 7c). The Rossby number, $Ro = u/fL$, where $u$ and $L$ are the mean velocity and width of the river inflow respectively, is $Ro \approx 10^3$ which indicates that the Coriolis force can be neglected. This corresponds to the observed weak Coriolis force ($10^{-5}$ ms$^{-2}$) relative to the other budget terms in the near-field region (Fig. 7c,g,k). The Coriolis acceleration becomes a primary balancing force in the mid and far-field plume regions (McCabe et al., 2009) and is generally a dominant factor in large-scale systems (Fong and Geyer, 2002).

Turbulence stress was surface-intensified, exceeding $\tau = 10^{-2}$ m$^2$s$^{-2}$ at the base of the plume (Fig. 7d). This maximum value is over one order of magnitude greater than peak $\tau$ within the Columbia River (Kilcher et al., 2012). The positive stress in the surface $2$ m indicates the movement of momentum downwards from the near-surface, where $\tau$ was an order of magnitude weaker, to the base of the surface layer at $4$ m. Below the surface layer, $\tau$ tended towards zero with depth. This resulted in large stress divergence within the surface layer about the base of the plume, and a decrease to a relatively steady $\partial \tau / \partial z$ below $4$ m (Fig. 7c).

Further downstream ($1 \leq x < 2$ km) the total deceleration was principally within the surface layer (Fig. 7g), driven by the along-channel advection component and local acceleration (Fig. 7f). Below the plume, $Du/Dt$ tended towards zero. The pressure gradient was approximately half the magnitude of $Du/Dt$ within the surface layer and comparable to $Du/Dt$ below the plume. Maximum $\tau = 10^{-3}$ m$^2$s$^{-2}$ at the base of the plume at $1.8$ m was an order of magnitude greater than $\tau$ within the plume while. Below the surface layer, $\tau$ tended to zero and was relatively constant which agrees with the weak deceleration observed (Fig. 7g,h). The high interfacial stress and weak ambient $\tau$ resulted in large stress divergence within the surface layer and a decrease in $\partial \tau / \partial z$ towards zero below $3$ m (Fig. 7g).

Towards the seaward end of Deep Cove ($2 \leq x \leq 3$ km) the along-channel advection component remained strong and dominated the near-surface deceleration (Fig. 7j). Below $4$ m, the weaker advection and spreading components balanced $\partial u / \partial t$ which led to a relatively steady ambient (Fig. 7k). The maximum pressure gradient was an order of magnitude less than $Du/Dt$ and relatively constant with depth. The stress at $z = 0$ was comparable to the maximum $\tau$ at the base of the plume (Fig. 7l), indicating that wind may have contributed a force to the surface layer at the seaward end of Deep Cove. This resulted in a surface-intensified stress divergence that reduced to zero with depth (Fig. 7k).

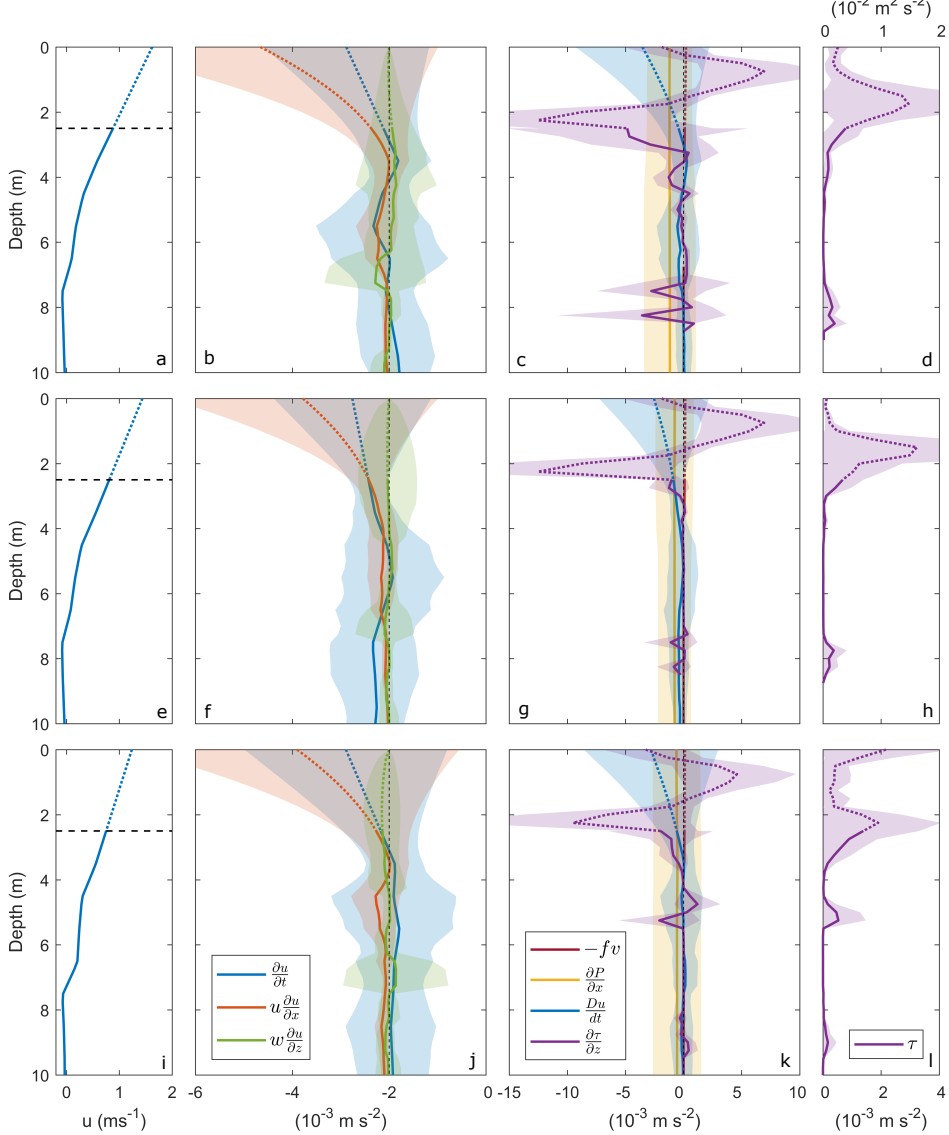

**Figure 7.** Terms in the along-channel momentum budget averaged over $0 \le x < 1$ km (top row), $1 \le x < 2$ km (middle) and $2 \le x < 3$ km (bottom). Profiles of (a,e,i) along-channel velocity $u$ (dashed lines indicate depth over which velocity was interpolated to the surface), (b,f,j) local acceleration (blue) and advection (x and z-component, orange and green respectively), (c,g,k) Coriolis force (red), pressure gradient (yellow), total acceleration (blue) and turbulence stress divergence (purple), and (d,h,l) internal turbulence stress ($\tau$, purple). The shading around each profile represents one standard deviation. The dotted profile above 2.5 m for each term indicates the region over which each variable was calculated using interpolated velocity data.

## 4 Discussion

### 4.1 Comparison of tailrace inflow with river discharges

The buoyant discharge into the head of Deep Cove is comparable to inflows of other coastal river plumes in many ways. The surface plume flowed at speeds greater than $1.5\ \mathrm{ms^{-1}}$ (Fig. 5b) which is consistent with the high flow speeds of the Columbia (Kilcher et al., 2012) and the Fraser Rivers (Tedford et al., 2009). However, the discharge rates of these other plumes are over one order of magnitude greater than the maximum $Q = 550\ \mathrm{m^3 s^{-1}}$ in Deep Cove. High inflows of $Q = 17,000$ and $10,000\ \mathrm{m^3 s^{-1}}$ have been recorded for the Columbia (McCabe et al., 2009) and Fraser Rivers (MacDonald and Horner-Devine, 2008) respectively. Studies of river plumes of a comparable discharge rate to the tailrace inflow, such as the Merrimack and Connecticut River, exhibit mean flow rates which are generally half the speed of the plume in Deep Cove (O'Donnell et al., 2008; Chen et al., 2009).

The continuous freshwater inflow to Deep Cove and substantial annual rainfall produced a highly stratified surface layer throughout the fjord. The maximum stratification at the base of the plume ($N^2 = 10^{-1}\ \mathrm{s^{-2}}$) (Fig. 5c) was approximately one order of magnitude greater than $N^2$ observed in the Columbia River during periods of large freshwater flux (Kilcher et al., 2012). Within the near-field region in Deep Cove, the highly stratified interfacial layer supported the intense shear generated by the high plume speeds, resulting in surface-intensified turbulence ($\epsilon > 10^{-3}\ \mathrm{Wkg^{-1}}$) (Fig. 6e) at least one order of magnitude larger than maximum $\epsilon$ measured in other river plumes of a comparable size (MacDonald et al., 2007; O'Donnell et al., 2008). The mean $\epsilon$ in the near-field of Deep Cove is instead comparable to the highest values recorded in other much larger river plume settings. Both the Fraser and Columbia Rivers displayed maximum $N^2 = 10^{-2}\ \mathrm{s^{-2}}$ at the base of their respective surface layers and the velocity shear across the pycnoclines resulted in peak $\epsilon > 10^{-4}\ \mathrm{Wkg^{-1}}$ (MacDonald and Geyer, 2004; Kilcher et al., 2012). Therefore, the tailrace discharge into Deep Cove exhibits flow properties that are comparable to much larger river systems and the near-field region of Deep Cove is amongst the most strongly stratified and turbulent in coastal regimes.

### 4.2 The balance of the momentum budget

The vertical structure of the water column and the momentum budget components vary significantly between the distinct layers of the upper water column (i.e., the surface plume, shear-stratified interface and ambient below) (Fig. 5, 7). Therefore, the balance of the components of Eq. (3), as illustrated in Fig. 8, were examined within each of these layers. The sum of the momentum budget terms averaged over the surface plume layer was approximately zero (Fig. 9d, pink) therefore, the budget was closed and a balance was achieved within the plume. The dynamical balance existed between $Du/Dt$ and $\partial\tau/\partial z$, where $Du/Dt$ was dominated by along-channel advection with a further, albeit weaker, contribution by the local acceleration term (Fig. 9a). The near-surface along-channel advection was greatest at the tailrace discharge point ($10^{-3}\ \mathrm{ms^{-2}}$) where maximum plume speeds were observed and generally decreased with distance, tending towards zero as flow speeds decreased (Fig 6c). The pressure gradient, approximately half the magnitude of the along-channel advection component, also acted to increase the total deceleration to balance the internal stress divergence. The overall balance between plume deceleration and turbulent

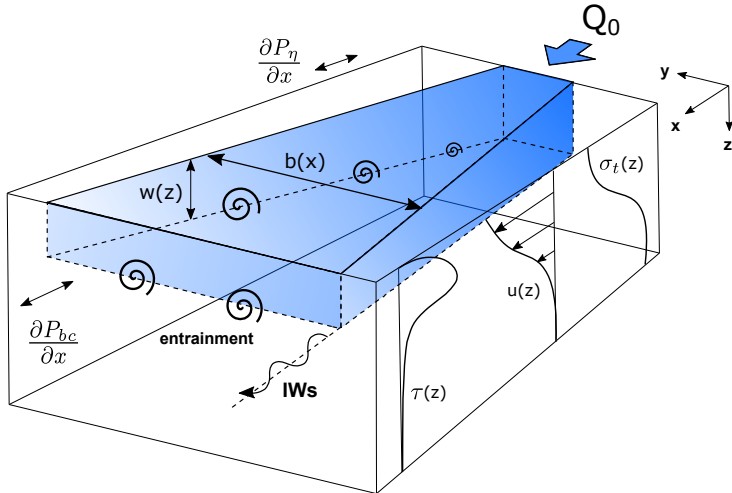

**Figure 8.** Schematic representation of the terms in the along-channel plume momentum budget, including near-field vertical structure. The plume is the blue shaded box fed by inflow $Q_0$. The increasing width of the plume in the across-channel direction is $b$ and the decreasing thickness of the plume is $h$. Vertical profiles of sigma-t ($\sigma_t$), along-channel velocity ($u$) and turbulence stress ($\tau$) are shown. The baroclinic ($P_{bc}$) and barotropic ($P_\eta$) components of pressure and the vertical entrainment velocity ($w$) are indicated. Entrainment between the plume and ambient is illustrated and internal waves (IWs) propagate along the base of the pycnocline.

stress divergence over the length of Deep Cove signifies that the vertical flux of low momentum, dense water into the freshwater surface layer controlled the deceleration of the river plume from the tailrace discharge point to as far as 3 km downstream. This balance of terms, where shear-driven turbulence acted as the primary driver of plume deceleration, has been previously observed in the near-field region of river plumes elsewhere (McCabe et al., 2009; Kilcher et al., 2012).

In the quiescent ambient below the surface layer ($h \leq z < 10$ m), a balance of momentum was also achieved (Fig. 9d, light blue). The momentum budget was dominated by the pressure gradient and the advection term without rotation, which is almost negligible (Fig. 9c); i.e., the Bernoulli principle. The spreading component was approximately half the magnitude of the along-channel advection term in the ambient and contributed to the total deceleration. Between $4 - 10$ m, $\tau$ was relatively weak and constant ($10^{-5}$ ms$^{-2}$) (Fig. 7d), linked to the reduced turbulent mixing at these depths (Fig. 5e), thus the stress divergence became negligible in the ambient (Fig. 9c).

However, the budget components averaged over the shear-stratified interfacial layer ($2 \leq z < h$ m) show that an along-channel momentum balance was not obtained within the interface over the inital 1 km (Fig. 9d, dark blue). The sum of the budget terms was negative as a result of weak deceleration ($Du/Dt < 0$) and a strong negative stress divergence term (Fig. 9b). In the initial 1 km downstream of the discharge point, there was no positive budget component within the interface and the total momentum exceeded $-5 \times 10^{-3}$ ms$^{-2}$ before tending towards zero farther downstream. This discrepancy in the budget

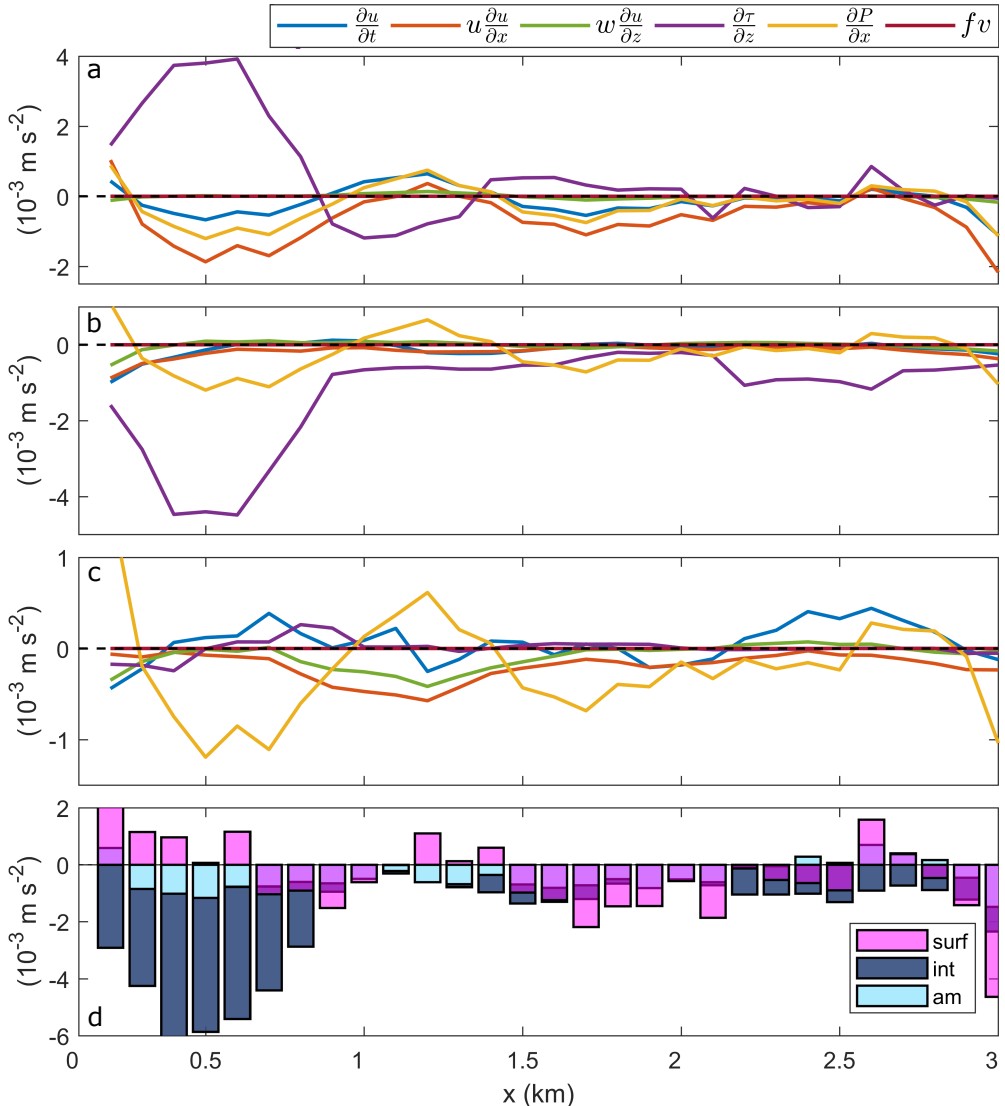

**Figure 9.** The individual and sum of all the layer-averaged terms in the along-channel momentum budget. The budget terms were averaged over the depth of (a) the plume ($0 \leq z < 2$ m), (b) the interface ($2 \leq z < h$ m), where $h$ is defined in Fig. 6b, and (c) the ambient below the surface layer ($h \leq z < 10$ m). (d) The sum of all the layer-averaged momentum components within the surface (pink) and interfacial (dark blue) layers, and the ambient (light blue). Horizontal dashed lines indicate zero, where the budget is completely balanced.

signifies that the input of momentum required to balance the output, primarily driven by turbulence, was not present within the interfacial layer.

### 4.2.1 Assessing the control volume accuracy

There are a number of possible sources for the discrepancy in budget terms in the shear-stratified interfacial layer (Fig. 9b,d). Firstly, that the budget components were not fully resolved by the observations, and measurement errors therein. Secondly, errors in the control volume technique arose from invalid assumptions. Thirdly, other processes, which were not accounted for in Eqn. 3, impacted the momentum balance within the interface of the Deep Cove system.

Potential under-sampling is a concern, particularly for the intermittent and heterogeneous turbulent field (MacDonald et al.,
2013) where peak $\epsilon$ occured within the upper 3 metres of the water column (Fig. 5d). However, as turbulence data was resolved right to the surface by the upwards-profiling microstructure profiler, stress was measured throughout the full vertical extent of the interfacial layer. Furthermore, as a balance of the momentum components was achieved within the surface layer (Fig. 9a,d), in which measurements are generally more difficult to resolve, insufficient observations seems an unlikely source of the discrepancy in the interface. Therefore, under-sampling should not greatly affect the stress divergence term in the interfacial
layer.

An alternate method of assessing the accuracy of the control-volume method and its estimates of the budget components is to compare the residual and directly measured internal stress divergence terms. When direct estimates of internal stress are unavailable, $\tau$ is defined as the residual in the momentum budget, i.e., the force required to balance the control volume estimate of total plume acceleration, pressure gradient and Coriolis force (MacDonald and Geyer, 2004). The residual internal stress
divergence was calculated within the plume, interface and ambient, and compared to the equivalent observed stress divergence in each layer. Both terms were averaged over each kilometer downstream of the tailrace discharge point to examine the along-channel difference between the two components, reflecting the spatial evolution of the other budget terms (Fig. 9). Turbulence dissipation rates were then derived from the residual stress divergence to determine the $\epsilon$ required to balance the momentum budget in the layer, and compared to the directly measured $\epsilon$. Within the surface plume and ambient below, there was generally
good agreement between the observed and residual internal stress divergence over the length of Deep Cove (Fig. 10). Both terms agreed within a factor of 2 which suggests that the control volume method produces a reasonable estimate of plume deceleration in these layers.

Within the shear-stratified interfacial layer, the internal stress divergence over the initial 1 km was overestimated by the control volume method (Fig. 10). While the observations show a strongly negative forcing ($-10^{-3}$ ms$^{-2}$), the residual indicates
a weakly positive value ($10^{-4}$ ms$^{-2}$) required to balance the momentum budget. The difference between the magnitude of stress divergence terms is a result of the underestimation of turbulence dissipation rates within the interface by the control volume method. The observed $\epsilon$ from the microstructure profiler in the initial 1 km are approximately one order of magnitude greater than the residual-derived $\epsilon = 10^{-4}$ Wkg$^{-1}$ estimates (Fig. 11), which leads to the weaker residual stress divergence. Downstream, both observed and residual $\epsilon$ compared well which was reflected in the good agreement between internal stress
divergence terms over 1 - 2 km and 2 - 3 km from the discharge point (Fig. 10).

The discrepancy in residual and observed $\epsilon$ in the interfacial layer (Fig. 11), thus internal stress divergence, was not due to the inability of the control volume method to resolve high dissipation rates. As $\epsilon$ was generally greatest within the surface layer

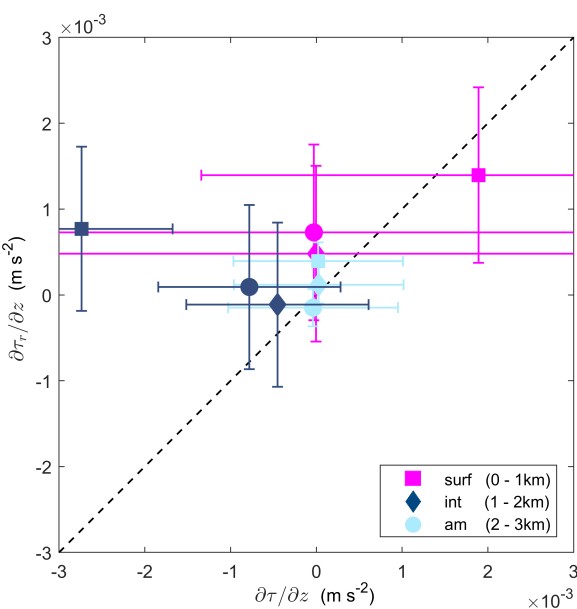

**Figure 10.** Directly observed internal stress divergence from microstructure profiler ($\partial \tau / \partial z$) compared to control-volume derived residual internal stress divergence ($\partial \tau_r / \partial z$). The terms are averaged over each layer: surface (pink), interface (dark blue), and ambient (light blue); and over each 1 km region from the tailrace discharge point: 0 - 1 km (square), 1 - 2 km (diamond), and 2 - 3 km (circle). The errorbars indicate one standard deviation for each term.

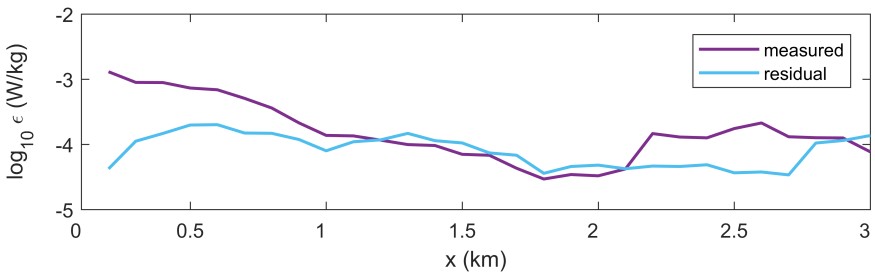

**Figure 11.** Comparison of interfacial layer-averaged $\epsilon$ over the length of Deep Cove, where $\epsilon$ was directly measured (purple) and derived from the residual stress divergence (blue).

and not the interfacial layer (Fig. 5d), the good agreement between the observed and residual stress divergence terms within the plume (Fig. 10, pink) illustrates that the control method can produce reasonable estimates of internal stress in plume systems

in which dissipation rates are enhanced. Control volume calculations of high turbulence stress have compared well to observed values in the near-field of the Columbia River, where $\epsilon > 10^{-4}$ Wkg$^{-1}$ (Kilcher et al., 2012).

    Furthermore, measurement error in the turbulence observations is also unlikely to be a major source of error to the control volume calculations. A thorough evaluation of the microstructure profiler sampling technique and the validity of measured $\epsilon$ in shear-stratified flows is conducted in the Appendix of McPherson et al. (2019). The angle of the microstructure profiler relative

the mean axial velocity increases as the VMP rises through the water column and meets the strong velocity gradients between the plume and ambient. However, the angle of the profiler did not generate erroneous $\epsilon$ when the tilt of the profiler was < 20 °C; the enhanced $\epsilon$ were instead representative of the intense shear-driven mixing in the surface layer. As the residual and observed dissipation rates within the interface compared well downstream of 1 km (Fig. 11), where the profiler continued to tilt due to the enhanced velocity shear throughout the whole near-field region, the angle of the profiler through the interface,

for $\theta_x < 20$ °C, was not responsible for the discrepancy between measured and residual $\epsilon$ estimates near the river mouth.

    The analysis above suggests that neither failing to fully resolve the budget components from observations, nor errors in the control volume method for estimating turbulence stress, are likely to be the main source of discrepancy between momentum budget components in the shear-stratified interfacial layer. Other potential errors in the control volume result from assumptions in estimates of lateral spreading, and assuming that the $v$-component of velocity was equal to zero along the plume axis. The

plume width is estimated from Eqn. 7 which provides a good estimate of lateral fluxes within the plume layer where $u$ and the salinity gradient are high (Kilcher et al., 2012), which is the case in the interfacial layer (Fig. 5). Furthermore, the control volume $b$ matched well with estimates of plume width derived from the GPS drifter experiment described in Section 2.2, and with measurements of $b$ from observed across-channel velocity structure (Fig. 1c). Thus error in the estimates of plume spreading should be minimal. The second assumption is that the plume is aligned with the along-channel transect (i.e., $v = 0$).

The sampling transects (Fig. 1c) suggest that the vessel was aligned with a plume streamline however, as the Coriolis term was non-zero (Fig. 7, 9), this indicates that sampling was not directly aligned with the core of the plume. While this could give an error in the control volume estimate of plume deceleration hence residual stress divergence, calculations in the surface layer are relatively unaffected by lateral effects due to the alignment of the streamwise direction with the mean plume flow direction. The general agreement between stress divergence terms within the surface plume (Fig. 10) indicates that any error introduced

by this assumption remains small. A more detailed consideration of these assumptions and their potential as sources of error to control volume calculations are discussued in Kilcher et al. (2012) and MacDonald and Geyer (2004), and future studies should more accurately resolve these terms to determine a more reasonable estimate of the control volume-derived stress component.

    The influence of other riverine physical processes, not resolved by the traditional budget, on the momentum of the system is now considered. To balance the strongly negative $\partial\tau/\partial z$ (Fig. 9b), a positive $Du/Dt$ is required. Return flows are intrinsic

to estuarine circulation and propagate in the opposite direction of the plume between the surface layer and ambient below (Pritchard, 1952). The up-fjord directed current would transport momentum back into the system along the pycnocline. In order to balance the observed $\partial\tau/\partial z = -10^{-3}$ ms$^{-2}$ within the interface, the return flow would have to increase by approximately 0.9 ms$^{-1}$ from the seaward end of Deep Cove to the tailrace discharge point. This is more than four times greater than the difference of $\sim 0.2$ ms$^{-1}$ between return flow velocities at either end of Deep Cove measured by O'Callaghan and Stevens

(2015). Therefore, a return flow would not be sufficient to contribute to the additional up-stream momentum required to balance the budget components in the interface.

## 4.3 The role of internal waves in the shear-stratified layer

The complex density interfaces and turbulence in the surface layer are clearly illustrated in the high intensity acoustic backscatter from the measurements by the shipboard echosounder (Fig. 4). The momentum within the near-surface plume is transferred
downwards by turbulence to the base of surface layer however, does not remain in the shear-stratified interfacial layer. The momentum is instead transferred away from the interface. The sounder shows internal waves propagating along the base of the surface layer which are capable of redistributing significant quantities of energy contained below the plume farther downstream (Nash and Moum, 2005).

The spectral distribution of temperature within the interfacial layer was inferred from the near-field mooring approximately
1 km downstream from the tailrace discharge point (Fig. 1c). The tidal signal, representing approximately 12% of the total energy in the signal, was filtered from the temperature data using classical harmonic analysis (T_TIDE, Pawlowicz et al. (2002)). The time series was then split into half-overlapping intervals equivalent to the inertial frequency and the spectrum was computed using Welch's periodogram method. The spectral fall-off rate of the continuum internal wave band power spectra varied with frequency (Fig. 12). A spectral slope of $\sigma^{-2}$ ('-2' on a log-log scale) was observed throughout the low-frequency
range of the internal wave band ($\sigma < 4f$ cpd) and transitioned to a slope of -5/3 for the higher frequency range ($10^2 < \sigma < N$ cpd). This transition to a weaker slope, instead of the signal following the steeper -2 slope from lower frequencies, indicates that there is a large amount of energy contained at these high frequencies, and the -5/3 slope suggests a mean dominance of shear-induced turbulence. The smooth spectral slope of -2 within the continuum internal wave band is consistent with the background canonical GM spectrum of internal waves in the open ocean (Garrett and Munk, 1972), which suggests that internal
waves could be one of the high-frequency processes that are captured by the energetic spectrum. The sounder transect illustrates their existence in this system. It is these internal waves which could be an unresolved process in the momentum budget, thus responsible for the balance within the shear-stratified interface not being closed using Eqn. 3 alone.

Internal waves were also observed propagating along the base of the interfacial layer at Elizabeth Island, approximately 5 km downstream of the tailrace discharge point (Fig. 13). The freshwater surface layer is relatively well-mixed and observed
as the dark backscatter from the water surface to approximately 1.5 m deep. The highly stratified ($N^2 > 0.4$ s$^{-2}$) base of the interfacial layer is evident as the dark band at 3 m depth. Below the interface, internal waves can be seen propagating at approximately 5 m depth. The echosounder transects in both the near-field (Fig. 4) and far-field plume region (Fig. 13), the transition between flow regimes (Fig. 6g), and the energetic high-frequency band of the temperature spectrum (Fig. 12), provide clear evidence of the existence of internal waves propagating along the shear-stratified interfacial layer.
Further evidence of the existence of internal waves is the observed transition from a supercritical ($Fr_i > 1$) to sub-critical ($Fr_i < 1$) flow regime at approximately 1 km downstream from the tailrace discharge point (Fig. 6g). The free propagation of internal waves during this change in flow regime have been identified at other plume fronts (Nash and Moum, 2005; Jay et al., 2009; Pan and Jay, 2009; Kilcher et al., 2010; Osadchiev, 2018). Illustrated as variability in the acoustic scattering along the

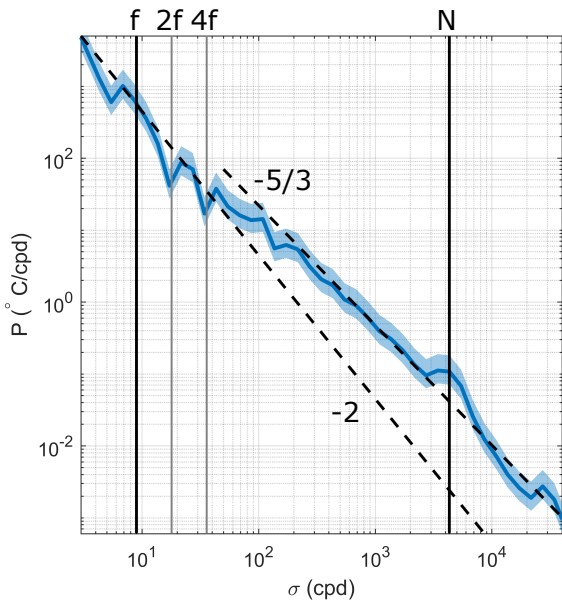

**Figure 12.** Spectral analysis of temperature in the interface (3.5 m) for the entire mooring time series, split into half-overlapping intervals equivalent to the inertial frequency (20 degrees of freedom) and the spectrum was computed using Welch's periodogram method. The focus is on the [f,N] range with particular frequencies highlighted. For reference, spectral slopes of -2 and -5/3 are indicated by the black dashed lines and a 95% confidence interval is the shaded region around the spectrum.

density interface in the sounder transect (Fig. 4), the variability in plume thickness and plume speed (Fig. 6b,c), which was not induced by a change in discharge rate (Fig. 2a), was also consistent with internal wave release (Kilcher et al., 2010). Note that the observed internal waves at $x = 0.8$ km in the echosounder transect (Fig. 4) occurred when near-surface plume speeds were $\sim 0.1$ ms$^{-1}$ weaker than when the along-channel transect in Fig. 6 was conducted. Therefore, the transition in Froude number would occur closer to the tailrace discharge point than when plume speeds were higher during the transect illustrated in Fig. 6g.

Strong stratification is favourable for the generation and propagation of internal waves and the $N^2$ observed here is amongst the highest recorded in the coastal ocean. The total vertical momentum flux divergence ($\partial F_z / \partial z$) of internal gravity waves was calculated, using Eqn. 2, to be $F_z = 1.75 \times 10^{-4}$ m$^2$s$^{-2}$. Thus the momentum flux divergence $\partial F_z / \partial z = 1.65 \times 10^{-4}$ ms$^{-2}$. This equates to approximately $14.9\%$ of the total measured momentum (Fig. 9) which suggests that, by removing energy along the pycnocline, internal waves could be an important mechanism of momentum transport within the shear-stratified interface.

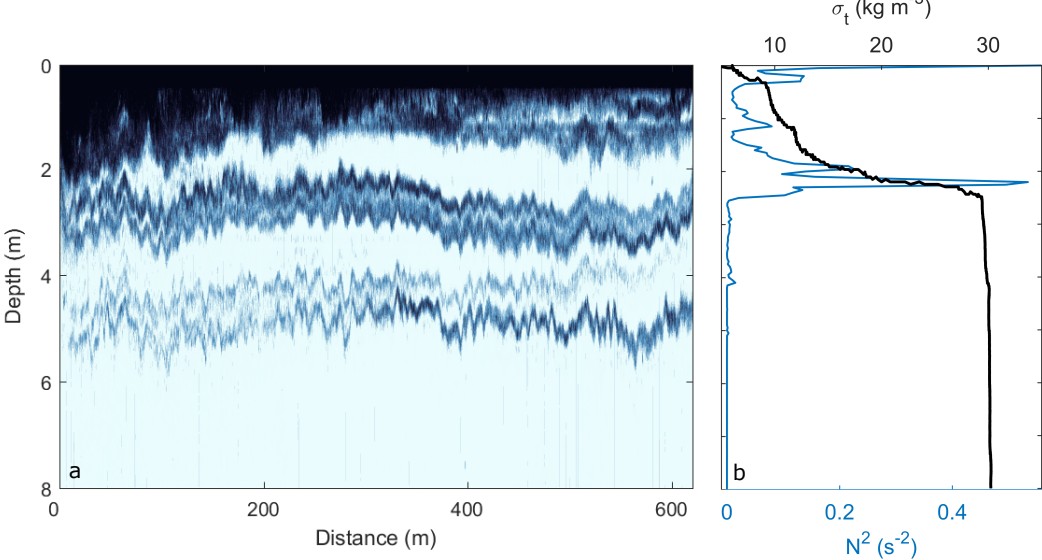

**Figure 13.** (a) Along-channel echosounder (EK60) transect from beside Elizabeth Island, with surface flow moving seaward from left to right and (b) vertical $\sigma - t$ (black) and buoyancy frequency-squared (blue) profiles from the VMP at the same location. Elizabeth Island is approximately 5 km downstream of the tailrace discharge point.

### 4.4 Momentum loss in the shear-stratified layer

As well as allowing for the release of internal waves, the transition from a supercritical to sub-critical flow regime can also indicate the presence of an internal hydraulic jump (Cummins et al., 2006). When flow dominated by kinetic energy (a super-critical flow) transitions into a flow dominated by its potential energy (a sub-critical flow), mechanical energy is released and is either radiated away by internal waves or dissipated locally (Nash and Moum, 2001; Klymak et al., 2004; Osadchiev, 2018). These jumps can alter the vertical structure of the stratified flow by intensifying density gradients, accelerating the flow and modifying vertical shear (Nash and Moum, 2001). Hydraulic jumps have previously been observed in Deep Cove, caused by variable discharge rates, as the fast surface plume discharged into the deep, stationary ambient presents an ideal environment for their generation (O'Callaghan and Stevens, 2015).

Observations of the evolving plume structure in the near-field region are suggestive of the presence of a hydraulic jump. Typically, jump occurence is corroborated by distinct differences in the vertical thermohaline structure and the transition in flow regime from super- to sub-critical. While both of these features were identified in the along-channel plume transect, a distinct and well-defined jump is not clearly evidenced in this data set. The clear decrease from $Fr_i > 1$ to $Fr_i < 1$ observed at approximately 1 km downstream of the discharge point (Fig. 6g) indicates a transition in flow regime. However, care should be taken when identifying a threshold value ($Fr_i = 1$) using the two-layer definition of $Fr_i$. This approximation in river plume

systems can accurately indicate changes in $Fr_i$ however, given the thickness of the shear interfacial layer (Fig. 5b), it is difficult to constrain the transition from supercritical to sub-critical flow, thus define a specific jump location.

A change in plume structure, characteristic of a hydraulic jump, also occured where the transition in $Fr_i$ was identified. The abrupt deceleration in flow speed from 1.2 to 0.8 $\mathrm{ms}^{-1}$ (Fig. 6c) and increase in plume thickness from 3.2 m to 5 m (Fig. 6b) is consistent with the thin, fast near-surface supercritical flow matched to the thicker, slower sub-critical layer. However, the high-frequency variability in the along-channel plume structure throughout the near-field region makes clearly identifying a potential hydraulic jump difficult. Such variability may instead represent changes in behaviour as the plume evolves. There exist limitations in observations with respect to hydraulic jumps due to the difficulties of resolving the sharp horizontal density and flow gradients, as well as their temporal evolution. Recently, a number of near-field plume studies have identified the formation of a hydraulic jump near river mouths using an extensive suite of remote sensing and in-situ observations (Honegger et al., 2017; Osadchiev, 2018). Thus such hydraulic mechanisms can exist in near-field plume systems under certain conditions however, the clear identification and constraining of such hydraulic processes with observational data remains challenging.

While characteristics of a hydraulic jump can be identified in the near-field plume, though not distinct and fully resolved, the potential influence of a hydraulic jump on the distribution of near-field momentum can still be speculated about. The power dissipated across a hydraulic jump can be estimated using $E = \rho g' Q \Delta H$, where $Q$ is the tailrace discharge rate and $\Delta H = (y_2 - y_1)^3 / 4 y_2 y_1$ (Weber, 2001). The depths for the surface plume and Deep Cove are $y_1 = 10$ m and $y_2 = 110$ m respectively, reflecting the large difference between the depth of the surface layer and the inner fjord, thus $\Delta H = 225$ m. When $Q = 530$ $\mathrm{m^3 s^{-1}}$, a total of $E \sim 28.7$ MW is dissipated across the jump. This is the equivalent to over 30 % of the total energy within the interface. A more conservative depth estimate for the supercritical and sub-critical layers of $y_1 = 2$ m and $y_2 = 10$ m respectively, as it is unlikely for the hydraulic jump to fill the entire water column, results in a total energy loss of 2 % across the hydraulic jump. Therefore, the hydraulic jump could contribute up to one third of the total dissipation of momentum within the interfacial layer and is thus a crucial, yet generally unconsidered, process in the balance of plume momentum.

## 5 Conclusions

The momentum budget constructed here using hydrographic and turbulence observations, integrated over a control volume, illustrates the role of each budget component on the distribution of momentum within the near-field region of a buoyant plume (Fig. 8). Using direct measurements of $\epsilon$ from the vertical microstructure profiles to calculate $\tau$, hence more accurately representing the mechanisms of turbulent mixing, is a development of the control volume technique established by MacDonald and Geyer (2004) which indirectly inferred $\tau$ from the residual of the budget components.

The influence of the budget components on near-field plume structure and evolution varied between the surface plume layer, the shear-stratified interfacial layer, and the quiescent ambient below. Within the plume, the turbulence stress divergence controlled the deceleration of the plume as far as 3 km downstream from the discharge point by entraining low-momentum ambient water into the surface layer; i.e., $Du/Dt$ balanced $\partial \tau / \partial z$ (Fig. 9a). The total plume deceleration and adverse pressure

gradient, approximately half the magnitude of $Du/Dt$, was also required to balance the stress divergence. This result directly confirms the control volume technique first applied by MacDonald and Geyer (2004) and validated by Kilcher et al. (2012).

The importance of internal stress as one of the dominant forces acting to decelerate the near-field plume has also been observed in other studies (McCabe et al., 2009; Kilcher et al., 2012).

Within the interfacial layer however, a balance of momentum using the budget terms was not achieved: there was no corresponding input of momentum to balance the output driven primarily by turbulence (Fig. 9b). This discrepancy in the momentum budget of the interfacial layer could be a result of the invalidity of control volume assumptions or other unresolved physical

processes in the near-field. Internal waves were observed propagating along the base of the surface layer, visible in both near and far-field plume regions (Fig. 4, 13), and were capable of transporting almost 15 % of the total energy out beyond the plume's boundaries. The generation of internal waves by river plumes and their transport of energy and momentum along the pycnocline has been previously observed in both large and small river systems (Nash and Moum, 2005; Pan and Jay, 2009; Osadchiev, 2018). Evidence of an internal hydraulic jump was suggested by a transition from a supercritical to sub-critical

flow regime in the initial 1 km (Fig. 6f) and a modification of plume flow speeds and vertical structure, characteristic of a hydraulic jump. However, the observations were unable to clearly resolve the sharp gradients and temporal evolution of a jump, thus the existence of such a hydraulic feature can only be speculated about. The momentum within the system which was not resolved by Eqn. 3 could be accounted for by considering the redistribution and dissipation of momentum by these dynamical processes. The consideration of internal hydraulics and wave radiation when evaluating a momentum budget in a

shear-stratified environment is therefore necessary to fully understand the impact of governing dynamics on plume behaviour and evolution.

It is anticipated that this work, which is directly applicable to the near-field plume region of a river entering the ambient ocean, can also be applied to other estuarine and coastal regimes, as well as stratified shear flows. The rapid river inflows into the coastal ocean can strongly influence both the structure and mixing dynamics of the plume, while the internal wave generation

and transports impact the physical, chemical and biological structure and processes in the larger scale coastal environment. Improved understanding of the dynamics and energetics in this system will enable better predictions of the ultimate fate of the freshwater discharge, materials and energy.

*Competing interests.* There are no competing interests present

*Acknowledgements.* The authors would like to thank Brett Grant, Mike Brewer, Tyler Hughen and June Marion who helped with the 2016

field experiments, and Meridian Energy for providing the tailrace flow data. We thank Alexander Osadchiev and an anonymous reviewer for their extensive comments which were very helpful in improving this manuscript. Bill Dickson and Sean Heseltine from the University of Otago skippered the vessels for the duration of the field campaign. This research was funded by the New Zealand Royal Society Marsden Fund, the Sustainable Seas National Science Challenge, and the National Institute of Water and Atmospheric Research Strategic Science

Investment Fund, and supported by the National Institute of Water and Atmospheric Research (NIWA). The data used are available at NIWA

Coastal and Marine Data Portal https://marinedata.niwa.co.nz/.

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
