# Peer review of "The role of turbulence and internal waves in the structure and evolution of a near-field river plume"

_Ocean Science, 2019_

## Referee Comment (RC1) · Alexander Osadchiev (Referee) · 10 Dec 2019

Review of "The role of turbulence and internal waves in the structure and evolution of a near-field river plume" by Rebecca McPherson et al. Submitted to: Ocean Science Manuscript number: 2019-120

Summary: In this paper the authors focus on structure and dynamics of a near-field part of the buoyant plume formed by the jet-like freshwater inflow with high velocity (> 2 m/s) and relatively small discharge rate (500-550 m3/s) into a deep and isolated fjord. The authors describe elaborate in situ measurements within the near-field plume and provide comprehensive analysis of the momentum budget of this complex dynamical

system based on the obtained data. They report several important features registered by in situ data at the river plume including anomalously high stratification, turbulence dissipation rate, and internal stress. They describe an internal hydraulic jump formed within the near-filed plume that generates energetic internal waves. The presented study evaluates the components of the momentum and energy budgets of this dynamical system and demonstrates the important role of internal waves in these budgets. The topic addressed in this manuscript and the obtained results are of great scientific and practical interest because similar processes are observed by satellite imagery in many world coastal areas where mountainous rivers inflow to sea and generate internal waves. Due to high quality and importance of the manuscript, I recommend this article to be published in Ocean Science after minor revision. Below I provide general comments and corrections that should be addressed by the authors.

1. One of the main drawbacks of this work is lack of in situ velocity measurements in the surface layer (top 2.5 m), which were linearly interpolated between the 0 m and 2.5 m measurements. However, the largest turbulence stress divergence was estimated to occur in this surface layer (Fig. 7), and these predicted values dominated the along-term momentum balance for the plume and the shear-stratified interfacial layer along the first 1 km of the transect (Fig. 9). Thus, usage of the linear interpolation for velocity in the top 2.5 m should be more thoroughly discussed and confirmed. This would provide a firm basis for the main results of the manuscipt. 2. No well-developed hydraulic jump was registered by in situ thermohaline or velocity measurements (e.g., Page 10, line 178-179). The hydraulic jump is predicted to form at a distance of 1 km from the freshwater inflow point (Fig. 6g), however, variability of the plume depth h at this part of the transect (between 3.5-4 and 5 m) was relatively low and did not exceed variability of h at the other part of the transect (Fig. 6b). Other characteristics of the plume also did not show any anomalous values near the predicted point of the hydraulic jump. Why the hydraulic jump was not detected by high-resolution in situ measurements? This issue should be addressed in the manuscript. 3. Internal waves generated by inflow of rivers at high speed to coastal sea are commonly visible at

satellite and aerial images. Is it the case of the internal waves generated in the Deep Cove? Did you analyze this kind of data? The paper might be strengthened by the related analysis. 4. Page 14, line 254. Why the depth of the plume was fixed equal to 2 m, while the depth of the shear-stratified interfacial layer was variable? It seems to be more appropriate to fix the depth of the shear-stratified interfacial layer and have variable plume depth. This point should be clarified. 5. Page 3, line 88 – page 4, line 89. Tidal amplitudes are relatively large, 1.5-2.5 m. What are values of tidal velocities? Do they influence mixing of the near-field part of the plume? 6. Page 9, line 173. Fig 5e -> Fig. 5d

---

## Referee Comment (RC2) · Anonymous Referee #2 · 18 Dec 2019

This manuscript presents the results of a momentum balance analyses in the near field region of an energetic fresh water plume entering into a fjord in New Zealand. Components of the momentum balance are derived directly from upward transiting microstructure profiles, and a modified version of the MacDonald and Geyer control volume analysis. Failure of the momentum balance to close, particularly within the interface region (2-4 m depth), is used as an opportunity to explore other mechanisms of energy dissipation, including forcing of internal waves, and energy dissipated through a hydraulic jump. These two missing mechanisms are identified as the missing elements in the mismatched momentum budget.

[Figure]

The authors present an interesting take on near field dynamics, and their suggested mechanisms for closing the momentum budget appear plausible. However, I have several concerns related to the data set, which lead me to some skepticism. The microstructure data set is discussed in more detail in McPherson et al (2019), particularly in Appendix A, where concerns about tilt angle and rate of change of the tilt angle of profile are disregarded. Given the significant ramifications of these extremely high dissipation measurements, both in this paper, and in McPherson et al (2019), I do not believe that the validity of these measurements has been thoroughly vetted. In McPherson et al (2019) these measurements are used to argue that Thorpe scale to Ozmidov scale ratios are several orders of magnitude below unity, in stark contrast to decades of observations in oceanic shear layers. In this paper, the measured dissipation values lead to dramatic conclusions about hydraulic jumps and internal waves, which, while plausible, are significant and groundbreaking. Given the potential significance of these two sets of conclusions and the fact that both are tied directly to dissipation measurements from microstructure profilers at extreme angles, I do not believe that the justification provided in the JGR paper is convincing. As one comparison, I suggest that the authors use the momentum balance in the interfacial layer to determine the stress divergence required to close the momentum budget, and compare that to the microstructure derived values, the Thorpe scale values of McPherson et al (2019) and other river plume environments. What dissipation rates would be required to close the budget? Without that context, it is difficult to gauge the relevance of the alternative mechanisms suggested by the authors. There is no doubt that these mechanisms may play a role to some degree, but their magnitude is at issue.

Additional comments (some minor) are as noted:

Line 112: The ADCP is also important in constraining $F_z$, correct?

Line 173: The referenced figure should be 5d

Line 179: $Fr_i$ is not defined, and a discussion should be included of exactly how the

quantity is calculated from the data. $Fr\_i$ is notoriously difficult to quantify in river plume environments due to the difficulty in constraining internal wave speed in a stratified (rather than two layer) flow. In using a classic two layer approximation, it is difficult (if not impossible) to accurately calculate a layer depth. As such, calculations of $Fr\_i$ in river plume environments typically have significant relative value (i.e., Fr is increasing or decreasing) but it is very difficult to constrain the crossing of supercritical to subcritical transitions. This becomes extremely important in later discussions regarding hydraulic jumps, so it is critical to back up these calculations here.

Figure(9): Panel (d) is extremely confusing, and does not appear to be consistent with panels (a), (b), and (c). For example, at 0.5 km, I would expect the pink bars (which are the sum of components in panel (a) to be of order 1 m/s2, the light blue bars to be of order 5 m/s2, and the blue bars to be of order 1-2 m/s2 (these are approximations by eye). The bars shown are not consistent with these approximations. Please clarify the intent of panel (d) or correct the plot if necessary.

Line 322: An increase in plume thickness from 3.3 to 5 m is suggested in Figure 6(b). While this jump does exist for two adjacent data points, a better interpretation (given the variability in h in 6b) might be a gradual increase in h from 4 to 4.3 m over approximately 1 – 2 km.

Line 330: The authors claim that hydraulic jumps are responsible for contributing up to 30% of the energy dissipation, based on entire water column calculations, which two lines earlier they suggest is unlikely. The 2% estimate is probably more realistic.

In summary, I strongly recommend that the authors further investigate the nature of the dissipation measurements from the microstructure profiler by comparing their measurements to budget derived estimates, and revisiting the extreme angle analysis and justification. This will provide essential context for further evaluation of the profiler data taken at extreme angles, and may have ramifications not only for the present manuscript but for the manuscript recently accepted by JGR-Oceans.

---

## Author Comment (AC2) · 19 Feb 2020

**Response to Reviewer 2 comments on McPherson et al., 'The role of turbulence and internal waves in the structure and evolution of a near-field river plume'– original comments and responses are headed in bold and italics respectively.**

**Reviewer Summary:**
This manuscript presents the results of a momentum balance analyses in the near-field region of an energetic fresh water plume entering into a fjord in New Zealand. Components of the momentum balance are derived directly from upward transiting

microstructure profiles, and a modified version of the MacDonald and Geyer control volume analysis. Failure of the momentum balance to close, particularly within the interface region (2 - 4 m depth), is used as an opportunity to explore other mechanisms of energy dissipation, including forcing of internal waves, and energy dissipated through a hydraulic jump. These two missing mechanisms are identified as the missing elements in the mismatched momentum budget.

*Response:*
We thank the Reviewer for their insightful and challenging comments and suggestions. We greatly appreciate the effort they have put into improving our work. Below, we have responded to all their comments.

**Reviewer Summary (ctd):**
The authors present an interesting take on near field dynamics, and their suggested mechanisms for closing the momentum budget appear plausible. However, I have several concerns related to the data set, which lead me to some skepticism. The microstructure data set is discussed in more detail in McPherson et al (2019), particularly in Appendix A, where concerns about tilt angle and rate of change of the tilt angle of profile are disregarded. Given the significant ramifications of these extremely high dissipation measurements, both in this paper, and in McPherson et al (2019), I do not believe that the validity of these measurements has been thoroughly vetted. In McPherson et al (2019) these measurements are used to argue that Thorpe scale to Ozmidov scale ratios are several orders of magnitude below unity, in stark contrast to decades of observations in oceanic shear layers.

*Response:*
We wish to be very clear - the results of McPherson et al. (2019) did not suggest that the ratio between the Thorpe ($L_T$) and Ozmidov ($L_O$) scales in shear-stratified flows

is flawed. As the Reviewer correctly points out, there are decades of observations which show the linear relationship between $L_T$ and $L_O$ in open ocean and coastal environments (e.g., Dillon, 1982; Ferron et al., 1998; Stansfield et al., 2001; Stevens, 2017; Wesson and Gregg, 1994). The results of McPherson et al. (2019) instead suggest that care must be taken when conducting turbulent overturn analysis to estimate dissipation rates using Thorpe scales in a stratified shear flow where boundary layers limit the size of turbulent overturns, and in turn restrict the derived maximum turbulence dissipation rates.

The Ozmidov scale expresses the vertical size an overturn can reach before affected by stratification. However, as $L_O$ is inferred and not directly measured, this length scale does not take into account either boundaries or variation in stratification. In the plume layer, where stratification is high ($N^2 = 10^{-2}$ s$^{-2}$), the inferred $L_O$ does not take into account that there is a free surface just above it, and below there is an order of magnitude increase in $N^2$ at the interface. Therefore, the length scale and the $L_O/L_T$ method are not flawed but the physical limitations of such vertical structure on fundamental length scales should be considered. Care should thus be taken when applying this method in shear-stratified flows where boundaries limit the size of overturns. This point is explicitly made in McPherson et al. (2019) to avoid this confusion, most notably in Sections 3, 5 and 6.

The impact of the tilt of the profiler ($\theta_x$), as well as the rate of change of tilt, on turbulence dissipation rates was thoroughly discussed in Appendix A of McPherson et al (2019). It was important to ascertain if the high dissipation rates were a result of the profiler tilting, causing errors in the calculation method, or if the tilting was a result of the high shear and stratification in the surface layer, which resulted in enhanced dissipation rates. By first examining profiler tilt, it was shown that increasing the tilt of the profiler relative to the vertical did not correspond to an increase in measured

Epsilon. When $\theta_x > 20°$ (the limit generally applied to profiler measurements, Lueck et al., 2013), the measured $\epsilon$ were at least one order of magnitude smaller than the maximum $\epsilon$ observed (Fig. A5a). The maximum dissipation rates were generally found within the surface layer (Fig. 9) and not at the interface where the measured tilt was greatest (Fig. A4b). There was also no correlation between the rate of change of tilt of the profiler and turbulence dissipation rates (Fig. A5b), as discussed in Appendix A. Thus, the high dissipation rates measured and the tilt of the profiler were due to enhanced turbulent mixing.

Furthermore, to ensure high quality data was used in the analysis, the data was filtered to remove all measurements when $\theta_x > 20°$. Thus, extremes of profiler tilt were not considered in either analysis of McPherson et al. (2019) or this manuscript. Additional evidence of the validity of observed turbulence dissipation rates within the interface is provided later in this Response, where $\epsilon$ derived from the residual stress divergence compared well with the observed $\epsilon$ (Response Fig. 1d).

We now included details about the method of calculation of turbulence dissipation rates from velocity shear and the thorough discussion of the microstructure data set in the manuscript.

*The VMP was deployed in an upwards-profiling mode, which enabled measurements right to the water surface. Due to contamination of data by the wake of the instrument, measurements of Epsilon towards the bottom of each profile were not always obtained. Due to the sharp velocity gradient between the fast-flowing surface plume and the quiescent ambient below, the instrument tilts relative to the vertical ($\theta_x$) as it rises. Data has been filtered to remove all measurements when $\theta_x > 20°$ (Lueck, 2013). Further details about the calculation of $\epsilon$ from velocity shear and other details pertaining to the microstructure data set, including the influence of profiler tilt on the calculation of*

*dissipation rates, are thoroughly discussed in Appendix A of McPherson et al. (2019).*

In closing this point about the validity of the microstructure data set, the Reviewer notes the high dissipation rates and compares the measured length scales with those observed in oceanic shear layers. However, it would have been more surprising if the results of McPherson et al. (2019) were exactly like an ocean shear layer because the systems and situations are so very different.

**Reviewer Comment:**
In this paper, the measured dissipation values lead to dramatic conclusions about hydraulic jumps and internal waves, which, while plausible, are significant and groundbreaking. Given the potential significance of these two sets of conclusions and the fact that both are tied directly to dissipation measurements from microstructure profilers at extreme angles, I do not believe that the justification provided in the JGR paper is convincing.

*Response:*
While the Reviewer exhibits skepticism about the microstructure data set, which we hope has been allayed following this response to the Reviewer, there are results presented in the manuscript not reliant on turbulence dissipation rates that also show evidence of internal waves and hydraulic jumps.

Firstly, the echosounder transects both in the near and far-field regions (Fig. 4, 11) clearly showed internal waves propagating along the base of the surface layer. The high spatial variability of plume thickness (Fig. 6a) also indicated these high-frequency internal waves (Osadchiev, 2018). The frequency spectra of temperature within the shear-stratified layer showed a -2 slope in the lower frequencies of the inertial

subrange (Fig. 10), consistent with the background canonical GM spectrum of internal waves (Garrett and Munk, 1972). Furthermore, the transition of Froude number from supercritical ($Fr > 1$) to sub-critical ($Fr < 1$) at approximately 1 km downstream of the tailrace discharge point (Fig. 6g) indicated the release of these observed internal waves. Other river plumes have been shown to generate internal waves as they transition from a super- to sub-critical state (Nash and Moum, 2005; Jay et al., 2009; Osadchiev, 2018).

The transition from supercritical to sub-critical flow can also induce a hydraulic jump (Weber, 2001). While the high-frequency variability in the along-channel plume structure was high, thus identifying clearly a hydraulic jump is difficult, the changes in plume structure at 1 km downstream are characteristic of a hydraulic jump. The increase in plume thickness by almost 2 m suggested the thinner supercritical flow matched to the thicker sub-critical flow, and the decrease of surface velocity by 0.4 m/s over the region also indicated the change in flow regime as a hydraulic jump is generated. It was in this downstream 1 km region that O' Callaghan and Stevens (2015) also observed a hydraulic jump, consistent with the results presented in this manuscript.

Broadly speaking, there is everything to suggest, in bulk terms, that the upper layer turbulence dissipation rate IS very high. So even if our unique dataset challenges the limits of technology, the scales that we resolve provide a provocative set of evidence for the community to work with. More detail about the observed characteristics of the hydraulic jump from the along-channel transect have been included in the manuscript:

*Hydraulic jumps have previously been observed in Deep Cove, caused by variable discharge rates, as the fast surface plume discharged into the deep, stationary ambient presents an ideal environment for their generation (O'Callaghan and Stevens, 2015).*

*These jumps can alter the vertical structure of the stratified flow by intensifying density gradients, accelerating the flow and modifying vertical shear (Nash and Moum, 2001). While variability in along-channel plume structure and behavior was high (Fig. 4, 6), thus identifying clearly a hydraulic jump is difficult, changes in the plume structure at 1 km downstream were characteristic of a hydraulic jump. At the jump location in Deep Cove, an increase in plume thickness from 3.2 m to 5 m (Fig. 6b) indicates where the thin near-surface supercritical flow matches the thicker sub-critical layer. While not anomalous, the sudden increase by almost 2 m is the largest change in h over the length of the fjord, and the plume continues to gradually thicken past 2 km downstream. A decrease in surface velocity from 1.2 to 0.8 m/s (Fig. 6c) indicates the abrupt deceleration of the fast super-critical flow as it transitions into a slower sub-critical flow and forms a hydraulic jump. Furthermore, jump-occurrence was corroborated by intense turbulence dissipation in the near-surface ($\epsilon > 10^{-3}$ W/kg) (Fig. 6e). When flow dominated by kinetic energy (a supercritical flow) transitions into a flow dominated by its potential energy (a sub-critical flow), mechanical energy is released and is either radiated away by internal waves or dissipated locally (Nash and Moum, 2001; Klymak et al., 2004; Osadchiev, 2018).*

**Reviewer Comment:**
As one comparison, I suggest that the authors use the momentum balance in the interfacial layer to determine the stress divergence required to close the momentum budget, and compare that to the microstructure derived values, the Thorpe scale values of McPherson et al (2019) and other river plume environments. What dissipation rates would be required to close the budget? Without that context, it is difficult to gauge the relevance of the alternative mechanisms suggested by the authors. There is no doubt that these mechanisms may play a role to some degree, but their magnitude is at issue.

*Response:*

The Reviewer makes a good suggestion here. Residual internal stress divergence (the sum of the total acceleration, pressure gradient and Coriolis force: a residual of the momentum budget) is generally used when direct measurements of stress are not possible, following MacDonald and Geyer (2004). A comparison between the residual and directly measured internal stress divergence terms is an alternate method of assessing the accuracy of the control-volume method. The authors have calculated the residual internal stress divergence and averaged it over each layer (the surface, interface and ambient), then compared it to the observed stress divergence term directly measured using the microstructure data set (see attached Response Fig. 1). The turbulence dissipation rates within the interface required to balance the budget were also derived from the residual stress divergence term, then compared to the directly measured dissipation rates from the microstructure profiler.

There was generally good agreement between the directly observed (purple) and residual (light blue) internal stress divergence terms within the plume layer (attached response Fig. 1a). The residual stress divergence in the initial 1 km downstream of the tailrace discharge point compared well to the high observed term ($> 3 \times 10^{-3}$ m/s$^2$), and both values, of a comparable order of magnitude, tended towards zero with distance downstream. This suggests that the control volume method produced a reasonable estimate of internal stress in the surface layer.

In the ambient below the plume, the residual internal stress divergence generally overestimated the observed term. The residual term was larger and exhibited greater variability than the directly measured term, which weakly contributed to the balance of momentum (Response Fig. 1c). This suggests that the under-sampling of turbulence observations below the plume (due to contamination of data by the wake of the instrument, measurements of Eps towards the bottom of each profile were not always obtained, see Fig. 5d) failed to resolve the turbulent field in the ambient. The discrepancy between the residual and observed terms however, is relatively small ($< 10^{-3}$ m/s$^2$) and the budget in the ambient using the observed internal stress divergence was relatively well-balanced (Fig. 9d).

In the interfacial layer over the initial 1 km, the directly measured internal stress divergence term did not compare well with the residual term (Response Fig. 1b). The estimated residual term was relatively constant at $1 \times 10^{-3}$ m/s$^2$ while the observed term sharply peaked at $-4 \times 10^{-3}$ m/s$^2$. The difference in the size of the residual and observed stress divergence terms within the interfacial layer is due to the underestimation of turbulence dissipation rates by the residual term. The observed $\epsilon$ in the initial 1 km are approximately one order of magnitude greater than the control-volume $\epsilon$ estimates (Response Fig. 1d), leading to the larger observed stress divergence term than residual term in the 0 - 1 km region (Response Fig. 1b). Downstream of 1 km, both estimates of dissipation rate compared well and both stress divergence terms were generally comparable and tended towards zero with distance.

The residual-derived turbulence dissipation rates close to the tailrace discharge point ($\epsilon = 10^{-4}$ W/Kg) are comparable to those observed in other river plumes of a comparable size to the tailrace inflow, such as the Connecticut (O'Donnel et al, 2008), the Hudson (Peters and Bokhorst, 2000) and the Merrimack River (MacDonald et al, 2007). The observed turbulence dissipation rates $\epsilon > 10^{-4}$ W/Kg) are comparable to those measured in much larger river systems like the Columbia River (Nash et al, 2009). A full comparison of $\epsilon$ in Deep Cove and other plume systems was conducted in Section 5.1 of McPherson et al. (2019).

The enhanced dissipation rates did not influence the validity of the control-volume method. As turbulence dissipation rates within the surface layer of Deep Cove were generally the greatest in the upper water column (Fig. 5d), the good agreement

between the stress divergence terms in the surface layer (Response Fig 1a) showed that the control method can produce reasonable estimates of stress in plume systems in which dissipation rates are enhanced. Therefore, the discrepancy in residual and observed turbulence dissipation rates in the interfacial layer, thus stress divergence terms, was not due to the control volume being unable to resolve high dissipation rates ($\epsilon$ is generally higher in the surface layer than the interface, Fig. 9 in McPherson et al, 2019). MacDonald et al. (2013) also found that the control-volume method can provide robust estimates of $\epsilon$ in a turbulent ($\epsilon < 10^{-4}$ W/Kg) near-field river plume.

Furthermore, the angle of the profiler also did not affect the dissipation rates observed. The tilt of the profiler as it rose through the sharp velocity gradient between the fast-flowing surface plume and the quiescent ambient below, about which the Reviewer expressed concern, was enhanced throughout the whole near-field region of Deep Cove. Large angles of tilt ($\theta_x > 20°$) were also found downstream of the 1 km region (Fig A4, A5 in McPherson et al, 2019). Therefore, as the dissipation rates within the interface derived from residual stress compared well to the microstructure estimates downstream of 1 km (Response Fig. 1d), where the profiler continued to tilt, this shows that the tilt did not influence the observed dissipation rates in the interface.

The authors therefore hypothesized that the residual stress divergence term instead did not resolve other turbulent processes that contributed to the larger observed dissipation rates in the interfacial layer of the initial 1 km (Response Fig. 1d). These processes were suggested as being internal waves and a hydraulic jump. Evidence of these processes using both the microstructure data and other instruments was presented in the manuscript (Fig 4, 6, 10, 11 and Sections 4.3 and 4.4), and further discussed in the Reviewer comment above.
**Additional Reviewer Comment:**
Line 112: The ADCP is also important in constraining $F_z$, correct?

*Response:*
The Reviewer is correct and this section has been moved to where the set-up of the ADCP is described in the methodology.

**Additional Reviewer Comment:**
Line 173: The referenced figure should be 5d

*Response:*
Corrected.

**Additional Reviewer Comment:**
Line 179: $Fr_i$ is not defined, and a discussion should be included of exactly how the quantity is calculated from the data. $Fr_i$ is notoriously difficult to quantify in river plume environments due to the difficulty in constraining internal wave speed in a stratified (rather than two layer) flow. In using a classic two layer approximation, it is difficult (if not impossible) to accurately calculate a layer depth. As such, calculations of $Fr_i$ in river plume environments typically have significant relative value (i.e., $Fr$ is increasing or decreasing) but it is very difficult to constrain the crossing of supercritical to sub-critical transitions. This becomes extremely important in later discussions regarding hydraulic jumps, so it is critical to back up these calculations here.

*Response:*
The Reviewer makes a good point. The authors used the internal Froude number,

$Fr = u_f/c$, where $u_f$ is the surface flow velocity estimated from the ADCP, and $c = \sqrt{g'H}$, where $H$ is the thickness of the surface layer, also used in many other river plume studies (Halverson and Pawlowicz, 2008; Jay et al., 2009; Hetland, 2010; O'Callaghan and Stevens, 2015; Osadchiev, 2018). The thickness of the surface layer ($H$) was defined by the depth of maximum stratification (Fig 5a,c).

A second method of examining the hydraulics of the Deep Cove plume was the composite Froude number, $G = F_1^2 + F_2^2$, following Armi and Farmer (1986), then Mac-Donald and Geyer (2005) and Geyer et al. (2017) etc. Where $F_i^2 = u_i^2/(g'h_i), (i = 1, 2)$, and $F_1$ and $F_2$ are the upper and lower layer Froude numbers, $u_i$ are the layer-averaged velocities, and $h_i$ are the thicknesses of the layers. The composite Froude number was calculated from the same along-channel transect in Fig. 6, then compared to the internal Froude number (see Response Fig. 2).

The results show that most of the contribution to $G$ comes from $F_1$ (Response Fig. 2) as the near-surface velocities are much greater than the relatively quiescent ambient (Fig. 5b). The value of $G$ is generally greater than $Fr_i$, though follows the same trend. Most importantly is that the location that $G$ transitions from super to sub-critical is approximately 1 km downstream of the discharge point, which is the same as where $Fr_i$ transitions from $Fr_i > 1$ to $Fr_i < 1$. Due to the agreement in these results, the authors are confident in the results showing a transition in Froude number in Fig. 6g using the internal Froude number.

Further details about the calculation of the Froude number have been included in the manuscript.

**Additional Reviewer Comment:**

Figure(9): Panel (d) is extremely confusing, and does not appear to be consistent with panels (a), (b), and (c). For example, at 0.5 km, I would expect the pink bars (which are the sum of components in panel (a) to be of order 1 m/s$^2$, the light blue bars to be of order 5 m/s$^2$, and the blue bars to be of order 1 - 2 m/s$^2$ (these are approximations by eye). The bars shown are not consistent with these approximations. Please clarify the intent of panel (d) or correct the plot if necessary.

*Response:*
While the panels above (Fig 5a – c) showed the dominant budget terms in each layer of the upper water column and their along-channel evolution respectively, it was difficult to determine, using these panels alone, if the budget across each layer was balanced. Therefore, Fig. 9d showed the sum of all the layer-averaged momentum budget terms in the surface (pink), the interface (dark blue) and ambient below (light blue) to illustrate the sum/balance of momentum in each layer, compared across all layers. The authors have amended the figure and caption to make the result clearer (see Response figure).

**Additional Reviewer Comment:**
Line 322: An increase in plume thickness from 3.3 to 5 m is suggested in Figure 6(b). While this jump does exist for two adjacent data points, a better interpretation (given the variability in h in 6b) might be a gradual increase in h from 4 to 4.3 m over approximately 1 – 2 km.

*Response:*
The Reviewer makes a good point. The variability and the gradual increase in plume thickness observed has been included and discussed in the manuscript (see comment above related to other observations of internal waves/hydraulic jumps for details).
**Additional Reviewer Comment:**
Line 330: The authors claim that hydraulic jumps are responsible for contributing up to 30% of the energy dissipation, based on entire water column calculations, which two lines earlier they suggest is unlikely. The 2% estimate is probably more realistic.

*Response:*
The estimate of 30% energy dissipation by hydraulic jumps was the maximum end of the range, calculated using the full water column values. However, the full range from 2 – 30% was always provided when discussing energy dissipation from jumps to give the Reader the full picture. These details, including the likelihood of this maximum estimate, was discussed in the text.

**Additional Reviewer Comment:**
In summary, I strongly recommend that the authors further investigate the nature of the dissipation measurements from the microstructure profiler by comparing their measurements to budget derived estimates, and revisiting the extreme angle analysis and justification. This will provide essential context for further evaluation of the profiler data taken at extreme angles, and may have ramifications not only for the present manuscript but for the manuscript recently accepted by JGR-Oceans.

*Response:*
We thank the Reviewer for their suggestions and recommendations. We hope our response serves to allay any concerns. The point about the tilting of the microstructure profiler is one that we were aware of from the outset, and do believe that we have taken sufficient care to account for potential issues.
* * *
**Fig. 1.** Momentum budget terms and residual stress divergence (light blue) averaged over (a) surface, (b) interface, (c) ambient. (d) (purple) observed and (blue) residual-derived interface dissipation rates

[Figure]

**Fig. 2.** Along-channel internal (Fr_i, black) and composite (G, yellow) Froude number, where G = F1^2 + F2^2, and F1 is the upper layer Froude number (blue) and F2 is the lower-layer Froude number (orange)

[Figure]

**Fig. 3.** Momentum budget terms averaged over (a) surface, (b) interface, (c) ambient layers. (d) The sum of the surface (pink), interface (dark blue) and ambient (light blue) layer-averaged components.

---

## Author Response (AR1)

**Response to Reviewer 1 comments on McPherson et al., 'The role of turbulence and internal waves in the structure and evolution of a near-field river plume'– original comments are in black, responses in blue.**

**Summary**:

In this paper the authors focus on structure and dynamics of a near-field part of the buoyant plume formed by the jet-like freshwater inflow with high velocity(>2 m/s) and relatively small discharge rate (500-550 m3/s) into a deep and isolated fjord. The authors describe elaborate in situ measurements within the near-field plume and provide comprehensive analysis of the momentum budget of this complex dynamical system based on the obtained data. They report several important features registered by in situ data at the river plume including anomalously high stratification, turbulence dissipation rate, and internal stress. They describe an internal hydraulic jump formed within the near-filed plume that generates energetic internal waves. The presented study evaluates the components of the momentum and energy budgets of this dynamical system and demonstrates the important role of internal waves in these budgets. The topic addressed in this manuscript and the obtained results are of great scientific and practical interest because similar processes are observed by satellite imagery in many world coastal areas where mountainous rivers inflow to sea and generate internal waves. Due to high quality and importance of the manuscript, I recommend this article to be published in Ocean Science after minor revision. Below I provide general comments and corrections that should be addressed by the authors.

The authors thank the Reviewer for their time and helpful suggestions for improvement of the manuscript. We were very pleased to see that they believe these results to be 'of great scientific and practical interest because similar processes are observed by satellite imagery in many world coastal areas where mountainous rivers inflow to the sea and generate internal waves'. Below, we have responded to their comments (in blue).

1.  One of the main drawbacks of this work is lack of in situ velocity measurements in the surface layer (top 2.5 m), which were linearly interpolated between the 0 m and 2.5 m measurements. However, the largest turbulence stress divergence was estimated to occur in this surface layer (Fig. 7), and these predicted values dominated the along-term momentum balance for the plume and the shear-stratified interfacial layer along the first 1 km of the transect (Fig. 9). Thus, usage of the linear interpolation for velocity in the top 2.5 m should be more thoroughly discussed and confirmed. This would provide a firm basis for the main results of the manuscipt.

This is an important point and we can provide the confirmation that the Reviewer seeks. In September 2015, a number of near-surface moorings were deployed in the initial 3km of in the near-field region of Doubtful Sound, which covers the same area as the control volume in this manuscript (Sept 2015 data presented in McPherson et al., 2019). The moorings consisted of high frequency upwards-facing ADCPs at 10m that measured velocity up to 1.25 m, and velocimeters measured velocity at approximately 0.2 m depth (Fig. 2 of McPherson et al., 2019). The velocity profiles from the base of the plume to 1.25 m were generally straight, and a linear fit to the velocity data to the surface was in excellent agreement with the velocimeter measurements at approx. 0.2 m (see figure 1 below).

[Figure]

*Fig 1: Linear interpolation of velocity profiles for high (orange) and low (blue) near-surface velocities at approx. 1 km downstream of tailrace discharge point. Velocity profiles from upwards-looking ADCP from 10 to 1.25 m (solid line), point velocity measurements from velocimeter at 0.2 m (stars), and interpolated velocity data to surface (dashed line).*

Similarly, the velocity profiles for the March 2016 data, on which this manuscript is based, from approx. 4.5 m to 2.5 m were generally straight (Fig. 5b in manuscript). Therefore, a linear fit to the VMADCP velocity data from 2.5 m to extrapolate velocity data to the surface was also applied. The interpolation to the surface also compared well to the velocity measurements derived from the surface drifters deployed at the tailrace discharge point and discussed in the text. Further discussion on these velocity measurements and methods were added to the manuscript, which now reads:

*Horizontal velocity estimates were obtained from a 600 kHz ADCP (RDI Workhorse) mounted on a pole alongside the vessel 1 m below the surface (Fig. 3). Currents were rotated according to the local bathymetry to determine along-channel (u) and across-channel (v) velocities. Velocity profiles were generally straight from the base of the plume to 2.5 m (Fig 5b), thus near-surface velocities were obtained by applying a linear fit to the velocity data to extrapolate from 2.5 m to the surface. The extrapolated velocity profile was in excellent agreement with velocity measurements from the velocimeters moored at 0.2 m. Near-surface velocity also compared well to surface currents derived from a series of Lagrangian GPS drifter experiments, in which a pack of surface drifters, released at the tailrace discharge point, were advected with the mean plume flow for approximately one hour (3 km). Furthermore, in-situ velocity measurements up to 1.25 m were obtained from previous field campaigns (McPherson et al., 2019), and good agreement was found between the linear fit of the extrapolated data at 1.25 m and the measured velocity in the surface layer.*

2.  No well-developed hydraulic jump was registered by in situ thermohaline or velocity measurements (e.g., Page 10, line 178-179). The hydraulic jump is predicted to form at a distance of 1 km from the freshwater inflow point (Fig. 6g), however, variability of the plume depth h at this part of the transect (between 3.5-4 and 5 m) was relatively low and did not exceed variability

of h at the other part of the transect (Fig. 6b). Other characteristics of the plume also did not show any anomalous values near the predicted point of the hydraulic jump. Why the hydraulic jump was not detected by high-resolution in situ measurements? This issue should be addressed in the manuscript.

The Reviewer correctly points out that there are no significantly anomalous values at the 1 km mark where the transition in Froude number occurs. The tailrace inflow itself is variable (Fig. 2a) and the variability in plume characteristics and vertical structure is high (Fig. 6) (O'Callaghan and Stevens, 2015, McPherson et al., 2019). This high variability in plume structure makes it difficult to identify clearly a hydraulic jump.

However, there are signals in the measurements at 1km that *are* characteristic of a hydraulic jump. The increase in plume thickness at 1 km by almost 2 m (Fig. 6b) was the largest change in plume thickness over the length of the fjord, and is consistent with the thin supercritical flow matching the thicker sub-critical layer. There was also a large decrease in surface velocity from 1.2 to 0.8 m/s (Fig. 6c), and the abrupt deceleration indicates the fast super-critical flow transitioning into a slower sub-critical flow and forming a hydraulic jump. Furthermore, while turbulence dissipation is enhanced throughout the near-field due to shear-stratified mixing (McPherson et al., 2019), a peak in Epsilon > $10^{-3}$ W/kg was observed at 1 km (Fig. 6e), suggestive of the intense turbulence generally observed within hydraulic jumps. These points are now clarified in the text, which reads:

*Hydraulic jumps have previously been observed in Deep Cove, caused by variable discharge rates, as the fast surface plume discharged into the deep, stationary ambient presents an ideal environment for their generation (O'Callaghan and Stevens, 2015). These jumps can alter the vertical structure of the stratified flow by intensifying density gradients, accelerating the flow and modifying vertical shear (Nash and Moum, 2001}. While variability in along-channel plume structure and behavior was high (Fig. 4, 6), thus identifying clearly a hydraulic jump is difficult, changes in the plume structure at 1 km downstream were characteristic of a hydraulic jump. At the jump location in Deep Cove, an increase in plume thickness from 3.2 m to 5 m (Fig. 6b) indicates where the thin near-surface supercritical flow matches the thicker sub-critical layer. While not anomalous, the sudden increase by almost 2 m is the largest change in h over the length of the fjord, and the plume continues to gradually thicken past 2 km downstream. A decrease in surface velocity from 1.2 to 0.8 m/s (Fig. 6c) indicates the abrupt deceleration of the fast super-critical flow as it transitions into a slower sub-critical flow and forms a hydraulic jump. Furthermore, jump-occurrence was corroborated by intense turbulence dissipation in the near-surface (Epsilon > $10^{-3}$ W/kg) (Fig. 6e). When flow dominated by kinetic energy (a supercritical flow) transitions into a flow dominated by its potential energy (a sub-critical flow), mechanical energy is released and is either radiated away by internal waves or dissipated locally (Nash and Moum, 2001; Klymak et al., 2004; Osadchiev, 2018).*

3. Internal waves generated by inflow of rivers at high speed to coastal sea are commonly visible at satellite and aerial images. Is it the case of the internal waves generated in the Deep Cove? Did you analyze this kind of data? The paper might be strengthened by the related analysis.

Due to the steep and narrow topography of Fiordland and Doubtful Sound, and extended periods of cloud cover (there are approximately 200 days/year of rainfall in Doubtful Sound), consistent and reliable satellite images of the plume are not available. However, internal waves propagating away from the plume were identified visually using a shore-mounted GoPro camera (see picture below). Though interesting qualitatively, using this data quantitatively to add to the current analysis is beyond the scope of this manuscript.

[Figure]

*Fig 2. Internal waves visually identified propagating away from the plume (screenshot from shore-mounted GoPro video)*

4.  Page 14, line 254. Why the depth of the plume was fixed equal to 2 m, while the depth of the shear-stratified interfacial layer was variable? It seems to be more appropriate to fix the depth of the shear-stratified interfacial layer and have variable plume depth. This point should be clarified.

While there was pronounced temporal and spatial variability in the vertical structure of the water column (Fig. 5, 6) (McPherson et al., 2019), stratification was used to delineate the distinct surface and interfacial layers from the ambient below. The depth of maximum $N^2$ best described the base of the interfacial layer (i.e., the base of the plume) (Fig 5a, c). The value of 2 m used to define the surface layer was the thickness of the surface layer in the mean density profile and, having reviewed the individual profiles, this definition of the surface layer was generally appropriate because it agreed with the thickness of the surface layer in a majority of density and stratification profiles. We did trial a threshold value of $N^2 = 0.1$ for the surface layer but less than 2% of the total measurements changed definition (from surface to interface, or vice versa). As the results did not vary significantly with this $N^2$ threshold, it suggests that for practical purposes the definition of the surface layer at 2 m is reasonable. Certainly the Reviewer's point would hold when looking at longer experiments where seasonal drivers would influence the 2 m scale.

5.  Page 3, line 88 – page 4, line 89. Tidal amplitudes are relatively large, 1.5-2.5 m. What are values of tidal velocities? Do they influence mixing of the near-field part of the plume?

This is a good point and was addressed in McPherson et al. (2019). The tidal signal was observed during steady tailrace discharge rates in the near-field region of Deep Cove by increases in plume thickness of approx. 0.2 m and velocity of approx. 0.2 m/s (Fig. 2a in McPherson et al., 2019). However, there was no correlation between estimates of turbulence dissipation rates over the surface layer and the tidal phase. Comparable maximum estimates of Epsilon were measured during both the ebb and flood tide in the near-field plume region (Figure 10b in McPherson et al., 2019). O'Callaghan and Stevens (2015) noted that the headwaters of the fjord absorbed the momentum of

tidal oscillations, which would in turn reduce the impact of tides on turbulent mixing in the near field. The manuscript has been amended to address this question, and now reads:

*The tides are predominantly semidiurnal with ranges of 1.5 m and 2.5 m for neap and spring tides respectively (Walters2001) however, the headwaters of the fjord absorb the momentum of tidal oscillations (O'Callaghan and Stevens, 2015) thus tides do not influence near-field mixing (McPherson et al., 2019).*

6.  Page 9, line 173. Fig 5e -> Fig. 5d

Noted and changed.

**Response to Reviewer 2 comments on McPherson et al., 'The role of turbulence and internal waves in the structure and evolution of a near-field river plume'– original comments are in black, responses in blue.**

**Summary**:

This manuscript presents the results of a momentum balance analyses in the near-field region of an energetic fresh water plume entering into a fjord in New Zealand. Components of the momentum balance are derived directly from upward transiting microstructure profiles, and a modified version of the MacDonald and Geyer control volume analysis. Failure of the momentum balance to close, particularly within the interface region (2-4 m depth), is used as an opportunity to explore other mechanisms of energy dissipation, including forcing of internal waves, and energy dissipated through a hydraulic jump. These two missing mechanisms are identified as the missing elements in the mismatched momentum budget.

We thank the Reviewer for their insightful and challenging comments and suggestions. We greatly appreciate the effort they have put into improving our work. Below, we have responded to all their comments (in blue).

The authors present an interesting take on near field dynamics, and their suggested mechanisms for closing the momentum budget appear plausible. However, I have several concerns related to the data set, which lead me to some skepticism. The microstructure data set is discussed in more detail in McPherson et al (2019), particularly in Appendix A, where concerns about tilt angle and rate of change of the tilt angle of profile are disregarded. Given the significant ramifications of these extremely high dissipation measurements, both in this paper, and in McPherson et al (2019), I do not believe that the validity of these measurements has been thoroughly vetted. In McPherson et al (2019) these measurements are used to argue that Thorpe scale to Ozmidov scale ratios are several orders of magnitude below unity, in stark contrast to decades of observations in oceanic shear layers.

We wish to be very clear - the results of McPherson et al. (2019) did not suggest that the ratio between the Thorpe and Ozmidov scales in shear-stratified flows is flawed. As the Reviewer correctly points out, there are decades of observations which show the linear relationship between Lt and LO in open ocean and coastal environments (e.g., Dillon, 1982; Ferron et al., 1998; Stansfield et al., 2001; Stevens, 2017; Wesson & Gregg, 1994). The results of McPherson et al. (2019) instead suggest that care must be taken when conducting turbulent overturn analysis to estimate dissipation rates using Thorpe scales in a stratified shear flow where boundary layers limit the size of turbulent overturns, and in turn restrict the derived maximum turbulence dissipation rates.

The Ozmidov scale expresses the vertical size an overturn can reach before affected by stratification. However, as LO is inferred and not directly measured, this length scale does not take into account either boundaries or variation in stratification. In the plume layer, where stratification is high ($N^2 = 10^{-2}$ s$^{-2}$), the inferred LO does not take into account that there is a free surface just above it, and below there is an order of magnitude increase in $N^2$ at the interface. Therefore, the length scale and the LO/Lt method are not flawed but the physical limitations of such vertical structure on fundamental length scales should be considered. Care should thus be taken when applying this method in shear-stratified flows where boundaries limit the size of overturns. This point is explicitly made in McPherson et al. (2019) to avoid this confusion, most notably in Sections 3, 5 and 6.

The impact of the tilt of the profiler, as well as the rate of change of tilt, on turbulence dissipation rates was thoroughly discussed in Appendix A of McPherson et al (2019). It was important to ascertain if the high dissipation rates were a result of the profiler tilting, causing errors in the

calculation method, or if the tilting was a result of the high shear and stratification in the surface layer, which resulted in enhanced dissipation rates. By first examining profiler tilt, it was shown that increasing the tilt of the profiler relative to the vertical did not correspond to an increase in measured Epsilon. When tilt > 20° (the limit generally applied to profiler measurements, Lueck et al., 2013), the measured Epsilon were at least one order of magnitude smaller than the maximum Eps observed (Fig. A5a). The maximum dissipation rates were generally found within the surface layer (Fig. 9) and not at the interface where the measured tilt was greatest (Fig. A4b). There was also no correlation between the rate of change of tilt of the profiler and turbulence dissipation rates (Fig. A5b), as discussed in Appendix A. Thus, the high dissipation rates measured and the tilt of the profiler were due to enhanced turbulent mixing.

Furthermore, to ensure high quality data was used in the analysis, the data was filtered to remove all measurements when tilt > 20°. Thus, extremes of profiler tilt were not considered in either analysis of McPherson et al. (2019) or this manuscript. Additional evidence of the validity of observed turbulence dissipation rates within the interface is provided later in this Response, where Eps derived from the residual stress divergence compared well with the observed Eps (Response Fig. 1d).

We now included details about the method of calculation of turbulence dissipation rates from velocity shear and the thorough discussion of the microstructure data set in the manuscript.

*The VMP was deployed in an upwards-profiling mode, which enabled measurements right to the water surface. Due to contamination of data by the wake of the instrument, measurements of Epsilon towards the bottom of each profile were not always obtained. Due to the sharp velocity gradient between the fast-flowing surface plume and the quiescent ambient below, the instrument tilts relative to the vertical (theta_x) as it rises. Data has been filtered to remove all measurements when theta_x > 20° (Lueck, 2013}. Further details about the calculation of Eps from velocity shear and other details pertaining to the microstructure data set, including the influence of profiler tilt on the calculation of dissipation rates, are thoroughly discussed in Appendix A of McPherson et al. (2019).*

In closing this point about the validity of the microstructure data set, the Reviewer notes the high dissipation rates and compares the measured length scales with those observed in oceanic shear layers. However, it would have been more surprising if the results of McPherson et al. (2019) were exactly like an ocean shear layer because the systems and situations are so very different.

In this paper, the measured dissipation values lead to dramatic conclusions about hydraulic jumps and internal waves, which, while plausible, are significant and groundbreaking. Given the potential significance of these two sets of conclusions and the fact that both are tied directly to dissipation measurements from microstructure profilers at extreme angles, I do not believe that the justification provided in the JGR paper is convincing.

While the Reviewer exhibits skepticism about the microstructure data set, which we hope has been allayed following this response to the Reviewer, there are results presented in the manuscript not reliant on turbulence dissipation rates that also show evidence of internal waves and hydraulic jumps.

Firstly, the echosounder transects both in the near and far-field regions (Fig. 4, 11) clearly showed internal waves propagating along the base of the surface layer. The high spatial variability of plume thickness (Fig. 6a) also indicated these high-frequency internal waves (Osadchiev, 2018). The frequency spectra of temperature within the shear-stratified layer showed a -2 slope in the lower frequencies of the inertial subrange (Fig. 10), consistent with the background canonical GM spectrum

of internal waves (Garrett and Munk, 1972). Furthermore, the transition of Froude number from supercritical (Fr > 1) to sub-critical (Fr < 1) at approximately 1 km downstream of the tailrace discharge point (Fig. 6g) indicated the release of these observed internal waves. Other river plumes have been shown to generate internal waves as they transition from a super- to sub-critical state (Nash and Moum, 2005; Jay et al., 2009, Osadchiev, 2018).

The transition from supercritical to sub-critical flow can also induce a hydraulic jump (Weber, 2001). While the high-frequency variability in the along-channel plume structure was high, thus identifying clearly a hydraulic jump is difficult, the changes in plume structure at 1 km downstream are characteristic of a hydraulic jump. The increase in plume thickness by almost 2 m suggested the thinner supercritical flow matched to the thicker sub-critical flow, and the decrease of surface velocity by 0.4 m/s over the region also indicated the change in flow regime as a hydraulic jump is generated. It was in this downstream 1 km region that O' Callaghan and Stevens (2015) also observed a hydraulic jump, consistent with the results presented in this manuscript.

Broadly speaking, there is everything to suggest, in bulk terms, that the upper layer turbulence dissipation rate IS very high. So even if our unique dataset challenges the limits of technology, the scales that we resolve provide a provocative set of evidence for the community to work with. More detail about the observed characteristics of the hydraulic jump from the along-channel transect have been included in the manuscript:

*Hydraulic jumps have previously been observed in Deep Cove, caused by variable discharge rates, as the fast surface plume discharged into the deep, stationary ambient presents an ideal environment for their generation (O'Callaghan and Stevens, 2015). These jumps can alter the vertical structure of the stratified flow by intensifying density gradients, accelerating the flow and modifying vertical shear (Nash and Moum, 2001}. While variability in along-channel plume structure and behavior was high (Fig. 4, 6), thus identifying clearly a hydraulic jump is difficult, changes in the plume structure at 1 km downstream were characteristic of a hydraulic jump. At the jump location in Deep Cove, an increase in plume thickness from 3.2 m to 5 m (Fig. 6b) indicates where the thin near-surface supercritical flow matches the thicker sub-critical layer. While not anomalous, the sudden increase by almost 2 m is the largest change in h over the length of the fjord, and the plume continues to gradually thicken past 2 km downstream. A decrease in surface velocity from 1.2 to 0.8 m/s (Fig. 6c) indicates the abrupt deceleration of the fast super-critical flow as it transitions into a slower sub-critical flow and forms a hydraulic jump. Furthermore, jump-occurrence was corroborated by intense turbulence dissipation in the near-surface (Epsilon > $10^{-3}$ W/kg) (Fig. 6e). When flow dominated by kinetic energy (a supercritical flow) transitions into a flow dominated by its potential energy (a sub-critical flow), mechanical energy is released and is either radiated away by internal waves or dissipated locally (Nash and Moum, 2001; Klymak et al., 2004; Osadchiev, 2018}.*

As one comparison, I suggest that the authors use the momentum balance in the interfacial layer to determine the stress divergence required to close the momentum budget, and compare that to the microstructure derived values, the Thorpe scale values of McPherson et al (2019) and other river plume environments. What dissipation rates would be required to close the budget? Without that context, it is difficult to gauge the relevance of the alternative mechanisms suggested by the authors. There is no doubt that these mechanisms may play a role to some degree, but their magnitude is at issue.

The Reviewer makes a good suggestion here. Residual internal stress divergence (the sum of the total acceleration, pressure gradient and Coriolis force: a residual of the momentum budget) is generally used when direct measurements of stress are not possible, following MacDonald and Geyer (2004). A comparison between the residual and directly measured internal stress divergence terms is an alternate method of assessing the accuracy of the control-volume method. The authors have

calculated the residual internal stress divergence and averaged it over each layer (the surface, interface and ambient), then compared it to the observed stress divergence term directly measured using the microstructure data set (see Response Fig. 1 below). The turbulence dissipation rates within the interface required to balance the budget were also derived from the residual stress divergence term, then compared to the directly measured dissipation rates from the microstructure profiler.

There was generally good agreement between the directly observed (purple) and residual (light blue) internal stress divergence terms within the plume layer (Response Fig. 1a). The residual stress divergence in the initial 1 km downstream of the tailrace discharge point compared well to the high observed term (> 3 x 10$^{-3}$ m/s$^2$), and both values, of a comparable order of magnitude, tended towards zero with distance downstream. This suggests that the control volume method produced a reasonable estimate of internal stress in the surface layer.

[Figure]

*Response Fig 1: The individual and sum of all the layer-averaged terms in the along-channel momentum budget, including the residual internal stress divergence term (light blue). The budget terms were averaged over the depth of (a) the plume (0 <= z < 2 m), (b) the interface (2 <= z < h m), where h is defined in Fig. 6b, and (c) the ambient below the surface layer (h <= z < 10 m). (d) The turbulence dissipation rates averaged over the interfacial layer, directly measured (purple) and derived from the residual internal stress divergence term (light blue).*

In the ambient below the plume, the residual internal stress divergence generally overestimated the observed term. The residual term was larger and exhibited greater variability than the directly measured term, which weakly contributed to the balance of momentum (Response Fig. 1c). This suggests that the under-sampling of turbulence observations below the plume (due to contamination of data by the wake of the instrument, measurements of Eps towards the bottom of each profile were not always obtained, see Fig 5d) failed to resolve the turbulent field in the ambient. The discrepancy between the residual and observed terms however, is relatively small ($< 10^{-3}$ m/s$^2$) and the budget in the ambient using the observed internal stress divergence was relatively well-balanced (Fig. 9d).

In the interfacial layer over the initial 1 km, the directly measured internal stress divergence term did not compare well with the residual term (Response Fig. 1b). The estimated residual term was relatively constant at $1 \times 10^{-3}$ m/s$^2$ while the observed term sharply peaked at $-4 \times 10^{-3}$ m/s$^2$. The difference in the size of the residual and observed stress divergence terms within the interfacial layer is due to the underestimation of turbulence dissipation rates by the residual term. The observed Eps in the initial 1 km are approximately one order of magnitude greater than the control-volume Eps estimates (Response Fig. 1d), leading to the larger observed stress divergence term than residual term in the 0 - 1km region (Response Fig. 1b). Downstream of 1 km, both estimates of dissipation rate compared well and both stress divergence terms were generally comparable and tended towards zero with distance.

The residual-derived turbulence dissipation rates close to the tailrace discharge point (Eps = $10^{-4}$ W/Kg) are comparable to those observed in other river plumes of a comparable size to the tailrace inflow, such as the Connecticut (O'Donnel et al, 2008), the Hudson (Peters and Bokhorst, 2000) and the Merrimack River (MacDonald et al, 2007). The observed turbulence dissipation rates (Eps > $10^{-4}$ W/Kg) are comparable to those measured in much larger river systems like the Columbia River (Nash et al, 2009). A full comparison of Eps in Deep Cove and other plume systems was conducted in Section 5.1 of McPherson et al. (2019).

The enhanced dissipation rates did not influence the validity of the control-volume method. As turbulence dissipation rates within the surface layer of Deep Cove were generally the greatest in the upper water column (Fig. 5d), the good agreement between the stress divergence terms in the surface layer (Response Fig 1a) showed that the control method can produce reasonable estimates of stress in plume systems in which dissipation rates are enhanced. Therefore, the discrepancy in residual and observed turbulence dissipation rates in the interfacial layer, thus stress divergence terms, was not due to the control volume being unable to resolve high dissipation rates (Eps is generally higher in the surface layer than the interface, Fig. 9 in McPherson et al, 2019). MacDonald et al. (2013) also found that the control-volume method can provide robust estimates of Eps in a turbulent (Eps < $10^{-4}$ W/Kg) near-field river plume.

Furthermore, the angle of the profiler also did not affect the dissipation rates observed. The tilt of the profiler as it rose through the sharp velocity gradient between the fast-flowing surface plume and the quiescent ambient below, about which the Reviewer expressed concern, was enhanced throughout the whole near-field region of Deep Cove. Large angles of tilt (theta_x > 20) were also found downstream of the 1 km region (Fig A4, A5 in McPherson et al, 2019). Therefore, as the dissipation rates within the interface derived from residual stress compared well to the microstructure estimates downstream of 1 km (Response Fig. 1d), where the profiler continued to tilt, this shows that the tilt did not influence the observed dissipation rates in the interface.

The authors therefore hypothesized that the residual stress divergence term instead did not resolve other turbulent processes that contributed to the larger observed dissipation rates in the interfacial layer of the initial 1 km (Response Fig. 1d). These processes were suggested as being internal waves and a hydraulic jump. Evidence of these processes using both the microstructure data and other

instruments was presented in the manuscript (Fig 4, 6, 10, 11 and Sections 4.3 and 4.4), and further discussed in the Reviewer comment above.

Additional comments (some minor) are as noted:
Line 112: The ADCP is also important in constraining F_z, correct?

The Reviewer is correct and this section has been moved to where the set-up of the ADCP is described in the methodology.

Line 173: The referenced figure should be 5d

Corrected.

Line 179: Fr_i is not defined, and a discussion should be included of exactly how the quantity is calculated from the data. Fr_i is notoriously difficult to quantify in river plume environments due to the difficulty in constraining internal wave speed in a stratified (rather than two layer) flow. In using a classic two layer approximation, it is difficult (if not impossible) to accurately calculate a layer depth. As such, calculations of Fr_i in river plume environments typically have significant relative value (i.e., Fr is increasing or decreasing) but it is very difficult to constrain the crossing of supercritical to subcritical transitions. This becomes extremely important in later discussions regarding hydraulic jumps, so it is critical to back up these calculations here.

The Reviewer makes a good point. The authors used the internal Froude number, Fr = u_f/c, where u_f is the surface flow velocity estimated from the ADCP, and c = sqrt(g'H), where H is the thickness of the surface layer, also used in many other river plume studies (Halverson and Pawlowicz, 2008; Jay et al., 2009; Hetland, 2010; O'Callaghan and Stevens, 2015; Osadchiev, 2018). The thickness of the surface layer (H) was defined by the depth of maximum stratification (Fig 5a,c).

A second method of examining the hydraulics of the Deep Cove plume was the composite Froude number, $G = F1^2 + F2^2$, following Armi and Farmer (1986), then MacDonald and Geyer (2005) and Geyer et al. (2017). Where $Fi^2 = ui^2/(g'hi)$, (i = 1,2), and F1 and F2 are the upper and lower layer Froude numbers, ui are the layer-averaged velocities, and hi are the thicknesses of the layers. The composite Froude number was calculated from the same along-channel transect in Fig. 6, then compared to the internal Froude number (see Response Fig. 2 below).

[Figure]

*Response Fig 2: Estimates of along-channel internal Froude number (Fri, black) and composite Froude number (G), where G = F1$^2$ + F2$^2$, and F1 is the upper-layer Froude number (blue) and F2 is the lower layer Froude number (orange).*

The results show that most of the contribution to G comes from F1 (Response Fig. 2) as the near-surface velocities are much greater than the relatively quiescent ambient (Fig. 5b). The value of G is generally greater than Fri, though follows the same trend.

Most importantly is that the location that G transitions from super to subcritical is approximately 1 km downstream of the discharge point, which is the same as where Fri transitions from > 1 to < 1. Due to the agreement in these results, the authors are confident in the results showing a transition in Froude number in Fig. 6g using the internal Froude number.

Further details about the calculation of the Froude number have been included in the manuscript.

Figure(9): Panel (d) is extremely confusing, and does not appear to be consistent with panels (a), (b), and (c). For example, at 0.5 km, I would expect the pink bars (which are the sum of components in panel (a) to be of order 1 m/s2, the light blue bars to be of order 5 m/s2, and the blue bars to be of order 1-2 m/s2 (these are approximations by eye). The bars shown are not consistent with these approximations. Please clarify the intent of panel (d) or correct the plot if necessary.

While the panels above (Fig 5a – c) showed the dominant budget terms in each layer of the upper water column and their along-channel evolution respectively, it was difficult to determine, using these panels alone, if the budget across each layer was balanced. Therefore, Fig. 9d showed the sum of all the layer-averaged momentum budget terms in the surface (pink), the interface (dark blue) and ambient below (light blue) to illustrate the sum/balance of momentum in each layer, compared across all layers. The authors have amended the figure and caption to make the result clearer (see figure below).

[Figure]

*Fig 9: The individual and sum of all the layer-averaged terms in the along-channel momentum budget. The budget terms were averaged over the depth of (a) the plume (0 <= z < 2 m), (b) the interface (2 <= z < h m), where h is defined in Fig. 6b, and (c) the ambient below the surface layer (h <= z < 10 m). (d) The sum of all the layer-averaged momentum components within the surface (pink) and interfacial (dark blue) layers, and the ambient (light blue). Horizontal dashed lines indicate zero, where the budget is completely balanced.*

Line 322: An increase in plume thickness from 3.3 to 5 m is suggested in Figure 6(b). While this jump does exist for two adjacent data points, a better interpretation (given the variability in h in 6b) might be a gradual increase in h from 4 to 4.3 m over approximately 1 − 2 km.

The Reviewer makes a good point. The variability and the gradual increase in plume thickness observed has been included and discussed in the manuscript (see comment above related to other observations of internal waves/hydraulic jumps for details).

Line 330: The authors claim that hydraulic jumps are responsible for contributing up to 30% of the energy dissipation, based on entire water column calculations, which two lines earlier they suggest is unlikely. The 2% estimate is probably more realistic.

The estimate of 30% energy dissipation by hydraulic jumps was the maximum end of the range, calculated using the full water column values. However, the full range from 2 − 30% was always provided when discussing energy dissipation from jumps to give the Reader the full picture. These details, including the likelihood of this maximum estimate, was discussed in the text.

In summary, I strongly recommend that the authors further investigate the nature of the dissipation measurements from the microstructure profiler by comparing their measurements to budget derived estimates, and revisiting the extreme angle analysis and justification. This will provide essential context for further evaluation of the profiler data taken at extreme angles, and may have ramifications not only for the present manuscript but for the manuscript recently accepted by JGR-Oceans.

We thank the Reviewer for their suggestions and recommendations. We hope our response serves to allay any concerns. The point about the tilting of the microstructure profiler is one that we were aware of from the outset, and do believe that we have taken sufficient care to account for potential issues.

---

## Author Response (AR2)

**Response to Reviewer 2 comments on McPherson et al., 'The role of turbulence and internal waves in the structure and evolution of a near-field river plume'– original comments and responses are headed in bold and italics respectively.**

**Author Comment:**
We thank the Reviewer for their insightful and helpful comments and suggestions. Their dedicated time and thoroughness have improved the quality of the manuscript. Below, we have responded to all their comments.

**Reviewer Comment:**
1. Concerns regarding tilt angle of VMP:
The authors have responded by pointing to the Appendix of the JGR paper to support their use of the VMP data set. I have conducted a careful review of the response as well as the earlier JGR paper, and the Lueck et al 2013 technical note. To a large degree, my concerns about the data set were motivated by concerns over the conclusion drawn in the JGR paper regarding the $L_T/L_O$ ratio, as best illustrated in Figure 12 of the JGR paper. Although this is not directly relevant to the paper currently under review, I am compelled to address these concerns here as they relate to the dissipation data set. Figure 12 (JGR) shows extremely high values of $L_O$, ranging from 1 m to 1000 km ($10^6$ m)! In my initial reading of the JGR paper I was not as focused on the range of values, but only that they were large, and thus attributed this to an over estimation of dissipation rates, which were measured at relatively high angles. Upon further inspection, there appears to be some other calculation error in the JGR manuscript affecting these values of Lo. Given the definition of Lo, and values of epsilon ranging from $10^{-6}$ to $10^{-2}$, appropriate values of N2 necessary to generate Lo values of 1000 km would be $10^{-12}$ to $10^{-9}$. Actual values of $N^2$ range from $10^{-3}$ to $10^{-1}$. Alternatively, note that the data shown in Figure 9(JGR) is associated with $L_O$ values falling between 1 cm and 1 m, using the axis values of $N^2$ and epsilon to calculate $L_O$. These values are very consistent with the range of values of $L_T$ shown in Figure 12 (JGR). Thus, my initial concerns regarding the dissipation data set may have been overstated, but I strongly encourage the authors to revisit the $L_O$ calculations relevant to the JGR paper and issue a correction as necessary. With regards to the shear probe data, I still have concerns about the highest dissipation values observed O($10^{-2}$ W/kg), which appear to be associated with attack angles exceeding 15 degrees. However, I recommend that these concerns be dealt with in the manuscript in comparison to, for example, the residual divergence estimates discussed further below, or that a cutoff angle of 10 or 15 degrees also be used and compared with the present averages. In closing on this topic, I should also mention that I agree entirely with the authors' comment that the measured dissipation rates should be much higher than those found in ocean shear layers. In fact, the observed values are largely consistent with values seen in the Columbia, the Merrimack,

and other plumes. My initial comment was not focused on the magnitude of the dissipation rates themselves in comparison to ocean shear layers, but only on the ratio of $L_T/L_O$. Given the points made above regarding $L_O$, this discrepancy may now be resolved.

*Response:*
The authors thank the Reviewer for their comments and the thoroughness of their response concerning the VMP data set in both papers. With regards to the shear probe data, a standard limit of $\theta = 20°$ was applied to the VMP data in both analyses as recommended by Osborn & Crawford (1980), and Lueck et al. (2013), amongst others. Reducing that limit to 10 or 15° would rule out the high dissipation rates that are representative of the strong velocity shear that actually drives the tilt. As concluded in the JGR Appendix, the high tilt of the profiler is a result of the strong shear which is accurately represented by the high $\epsilon$; the high $\epsilon$ are not erroneous signals due to the tilt. The issue of VMP tilt and the impact on $\epsilon$ is now addressed when examining the residual stress divergence in the revised manuscript in the new sub-section (Assessing the control volume accuracy). The relevant part of the section reads, with reference to figures in the manuscript, is below. The point about $L_O$ is worth further examination and highlights the nature of scaling. $L_O$ is not a "real" quantity in the sense of being something tangibly measurable, instead it is inferred. The range of $L_O$ in the JGR paper does suggest to us some further sampling is required, perhaps with a different platform. A moored profiler would be a suitable way of capturing the variability across this interface.

*The discrepancy in residual and observed $\epsilon$ in the interfacial layer (Fig. 11), thus internal stress divergence, was not due to the inability of the control volume method to resolve high dissipation rates. As $\epsilon$ was generally greatest within the surface layer and not the interfacial layer (Fig. 5d), the good agreement between the observed and residual stress divergence terms within the plume (Fig. 10, pink) illustrates that the control method can produce reasonable estimates of internal stress in plume systems in which dissipation rates are enhanced. Control volume calculations of high turbulence stress have compared well to observed values in the near-field of the Columbia River, where $\epsilon > 10^{-4}$ W kg$^{-1}$ (Kilcher et al., 2012).*

*Furthermore, measurement error in the turbulence observations is also unlikely to be a major source of error to the control volume calculations. A thorough evaluation of the microstructure profiler sampling technique and the validity of measured $\epsilon$ in shear-stratified flows is conducted in the Appendix of McPherson et al. (2019). The angle of the microstructure profiler relative the mean axial velocity increases as the VMP rises through the water column and meets the strong velocity gradients between the plume and ambient. However, the angle of the profiler did not generate erroneous $\epsilon$ when the tilt of the profiler was $< 20°$ C; the enhanced $\epsilon$ were instead representative of the intense shear-driven mixing in the surface layer. As the residual and observed dissipation rates within the interface compared well downstream of 1 km (Fig. 11), where the profiler con-*

*tinued to tilt due to the enhanced velocity shear throughout the whole near-field region, the angle of the profiler through the interface, for $\theta_x < 20° \ C$, was not responsible for the discrepancy between measured and residual $\epsilon$ estimates near the river mouth.*

**Editor Comment:**
The reviewer 2 detailed comments relevant to this manuscript are provided below. These provide some advise to consider in the final revision of the manuscript.

**Reviewer Comment:**
2. Internal waves and Hydraulic Jump:
The authors present strong evidence of internal waves observed at or below the interface, along with a reasonable estimate of their energy as a percentage of the total measured momentum. These calculations suggest that IWs may play a minor, but not negligible, role in the local dynamics. Evidence of a hydraulic jump is less clear from the available data. Changes in velocity and layer thickness are very difficult to discern from Figure 6, and may more realistically represent gradual changes as the plume evolves. Distinct and well defined hydraulic jumps are notoriously difficult to identify in these environments, and have remained elusive to many plume researchers. In the text added on hydraulic jumps, the authors point to the sharp difference in layer thickness and velocity at 1 km. Unfortunately, for both variables, the point at 1 km is an anomaly, and should not be considered in isolation. Consistent with this interpretation, the authors point to high dissipation rates in figure 6(e). These rates do not spike at the 1 km location, but are high throughout the early stages of the plume, consistent with rapid acceleration of the upper layer in the liftoff zone, followed by a gradual decrease beyond 1 km as the plume widens and decelerates. Thus, I believe the discussion about hydraulic jumps is not consistent with the data set and should be revised.

*Response:*
While hydraulic jumps are still discussed in the manuscript as the results do suggest their existence, the conclusions drawn from the data about the presence of jumps in Deep Cove have been revised. The conclusion has been been edited and the section which addresses hydraulic jumps and their existence in Deep Cove (Section 4.4) now reads:

[revised manuscript text omitted]

**Reviewer Comment:**
3. Stress divergence derived from the momentum balance:
This is important and useful information, and I appreciate the authors undertaking this analysis. I would strongly suggest that elements of the discussion included in the review be incorporated into the manuscript, including possibly Response Figure 1. However, the discrepancy between residual stress and measured stress in the interfacial layer (where tilts are presumably highest) is still troubling. The authors suggest that this demonstrates inadequacies in the residual method but do not explain why or how, except for pointing to internal waves and hydraulic jumps as possible energy sinks, but the manuscript and data are far from conclusive. Thus, it is unclear to me whether there are discrepancies in the residual method, or the measurements, or both. I suggest that this issue be tackled head on in the manuscript, even if there is no clear resolution.

*Response:*
A new section with this analysis and further elaboration about the potential sources for discrepancy, as the Reviewer outlines, is also included here (Section 4.2.1 Assessing the control volume accuracy), with new figures to better illustrate the analyses and determine sources of error (Figure 10 & 11). In this section, further discussion about the microstructure profiler tilt in the interface and its effects on measured $\epsilon$ are undertaken, as well as a deeper investigation of the control volume method and its assumptions. The aim is to better understand the potential inadequacies within the method and their extent, before suggesting alternate mechanisms for energy transfer. The section is also included here, with the new figures included in the revised manuscript:

[revised manuscript text omitted]

**Reviewer Comment:**

4. Additional comment regarding $Fr_i$:
The authors provide a clearer response regarding their definition of $Fr_i$, elements of which should be included in the manuscript. Note however, that $G^2 = F1^2 + F2^2$, which should put $G$ much more in line with $F1$, as is typical in plume environments. That said, the two layer approximation can be a good indicator of changes in Froude number, but given the thickness of the shear layer (and lack of two well defined layers), it should not be used to identify thresholds such as $Fr = 1$. Small changes in layer thickness can result in significant changes in $Fr$, and the profiles should only be taken to demonstrate changes in $Fr$ (e.g., sharp or gradual decreases after $\sim 0.5$ km).

*Response:*
This is an interesting point about the Froude number and how it can and should be used outside the laboratory where such two-layers systems are not always so distinct. The authors have amended the discussion to reflect this point and the conclusions drawn from the Froude number. The text is included in the above section (4.2.1). The definition of $Fr_i$ from the previous response has also been included in the methodology and now reads:

> *The internal Froude number, $Fr_i = u_f/c$ is defined using the vessel-based instrumentation, where $u_f$ is the near-surface flow velocity estimated from the ADCP, and $c = \sqrt{g'H}$, where $H$ is the thickness of the surface layer defined by the depth of maximum stratification. This definition of $Fr_i$ has been used in previous river plume studies to determine flow regimes (Hetland, 2010; O'Callaghan and Stevens, 2015; Osadchiev, 2018).*

**Reviewer Comment:**

5. Additional comment regarding plume thickness:
As discussed above, the new text still focuses primarily on the drop between two adjacent points, and is misleading.

*Response:*
The text has been amended to reflect the change in the focus of the discussion

surrounding hydraulic jumps, included in Section 4.2.1 above.

**Reviewer Comment:**
6. Additional comment regarding hydraulic jump contribution to the energy budget:
As discussed at length above, the data does not provide convincing evidence of a hydraulic jump. Researchers have been speculating for decades about the existence and/or importance of hydraulic jumps in plume dynamics, but no one has been able to successfully identify and constrain with data, the existence of these jumps. This study is no exception, as the data presented may be suggestive of a jump, but is full of ambiguities. A major driver in the lack of well defined hydraulic jumps in plume regions is likely the result of width expansion and stratified shear mixing driving deceleration of the upper layer, resulting in a "softer" transition to subcritical flow that occurs over several km, rather than a sharp and well defined hydraulic jump. Thus, I believe that the language regarding hydraulic jumps in the manuscript should be softened, and the manuscript should be revised to reflect the reality of, and limitations of, the observations with respect to hydraulic jumps.

*Response:*
The authors response to the topic of hydraulic jumps have been discussed in detail above. Pertaining to this point, the authors have amended sections of the manuscript when connecting internal waves, hydraulic jumps and the energy budget. The relevant part of the conclusion now reads:

*Internal waves were observed propagating along the base of the surface layer, visible in both near and far-field plume regions (Fig. 4, 13), and were capable of transporting almost 15 % of the total energy out beyond the plume's boundaries. The generation of internal waves by river plumes and their transport of energy and momentum along the pycnocline has been previously observed in both large and small river systems (Nash and Moum, 2005; Pan and Jay, 2009; Osadchiev, 2018). Evidence of an internal hydraulic jump was suggested by a transition from a supercritical to sub-critical flow regime in the initial 1 km (Fig. 6f) and a modification of plume flow speeds and vertical structure, characteristic of a hydraulic jump. However, the observations were unable to clearly resolve the sharp gradients and temporal evolution of a jump, thus the existence of such a hydraulic feature can only be speculated about. Thus the momentum within the system which was not resolved by Eqn. 3 could be accounted for by considering the redistribution and dissipation of momentum by these processes. The consideration of internal hydraulics and wave radiation when evaluating a momentum budget in a shear-stratified environment is therefore necessary to fully understand the impact of governing dynamics on plume behaviour and evolution.*

**Reviewer Comment:**

Overall, I still believe that the study contains an interesting data set, and that many of the calculations and analyses may be valuable to the community. However, the authors should be cautious in reaching too far beyond the data in drawing their conclusions.

*Response:*
The authors thank the Reviewer for their suggestions and recommendations, and recognition of the value of this analyses. We hope that our response has sufficiently addressed any concerns.